# Advances in Histological and Molecular Classification of Hepatocellular Carcinoma

**DOI:** 10.3390/biomedicines11092582

**Published:** 2023-09-20

**Authors:** Joon Hyuk Choi, Swan N. Thung

**Affiliations:** 1Department of Pathology, Yeungnam University College of Medicine, Daegu 42415, Republic of Korea; 2Department of Pathology, Molecular and Cell-Based Medicine, Icahn School of Medicine at Mount Sinai, 1468 Madison Avenue, New York, NY 10029, USA; swan.thung@mountsinai.org

**Keywords:** hepatocellular carcinoma, molecular genetics, classification, pathology, immunohistochemistry

## Abstract

Hepatocellular carcinoma (HCC) is a primary liver cancer characterized by hepatocellular differentiation. HCC is molecularly heterogeneous with a wide spectrum of histopathology. The prognosis of patients with HCC is generally poor, especially in those with advanced stages. HCC remains a diagnostic challenge for pathologists because of its morphological and phenotypic diversity. However, recent advances have enhanced our understanding of the molecular genetics and histological subtypes of HCC. Accurate diagnosis of HCC is important for patient management and prognosis. This review provides an update on HCC pathology, focusing on molecular genetics, histological subtypes, and diagnostic approaches.

## 1. Introduction

Hepatocellular carcinoma (HCC) is a primary liver malignancy composed of epithelial cells with hepatocellular differentiation and accounts for 75–85% of all primary liver cancers [1]. Primary liver cancer is the sixth most common cancer worldwide and the fourth most common cause of cancer-related deaths, with an estimated 841,080 new cases and 781,631 deaths in 2018. It is estimated that, by 2025, more than a million people will be affected by liver cancer per year [2].

Significant efforts have been made to characterize the molecular pathogenesis of HCC in recent years [3,4,5,6,7]. The landscape of molecular genetic alterations in HCCs was characterized using large-scale molecular profiling studies. Several subtypes of HCC with distinctive clinical, molecular, and histopathological features have been identified and outlined in the 2019 World Health Organization (WHO) classification of tumor series, 5th edition on digestive system tumors [1]. Additionally, the definition of HCC and its distinction from other entities were clarified in the updated classification [8]. Table 1 shows the evolution of the WHO classification of HCC.

Accurate pathological classification and understanding of the molecular pathogenesis of HCC are essential for patient management and prognosis. The diagnosis and treatment of HCC remain challenging owing to its histological and molecular genetic diversity. This review provides an update on HCC pathology with a focus on molecular genetics, histological subtypes, and diagnostic approaches.

## 2. Etiology

HCC frequently occurs in patients with chronic liver disease with cirrhosis (~80%), while approximately 20% of patients have non-cirrhotic liver [9,10]. Various etiologies have been linked to HCC development. The most common underlying chronic liver diseases are hepatitis B [11], hepatitis C [12,13], alcoholic liver disease, non-alcoholic fatty liver disease (e.g., metabolic syndrome) [14,15], and several inherited diseases (e.g., genetic hemochromatosis, glycogen storage diseases, and hereditary tyrosinemia). Exogenous exposure to fungal toxins (e.g., aflatoxin B1 produced by *Aspergillus* species) through food contamination is a critical cause in tropical and subtropical regions [16]. In rare cases, HCC results from malignant transformation of hepatocellular adenoma (HCA) [17].

## 3. Pathogenesis

The mechanisms of hepatocarcinogenesis can be divided into (1) etiology-specific and (2) non-specific mechanisms [18,19]. Etiology-specific mechanisms have been identified in HCC induced by hepatitis B, hepatitis C, and aflatoxin B1 exposure. Hepatitis B virus (HBV)-induced hepatocarcinogenesis involves complex processes, including (1) integration of the HBV genome into the host genome and the associated host DNA deletions in cancer-related genes (e.g., telomerase reverse transcriptase [*TERT*] promoter); (2) HBx transcriptional activation, which can alter the expression of cell growth regulating genes; (3) viral–endoplasmic reticulum interactions that induce oxidative stress; and (4) targeted activation of oncogenic signaling pathways by various HB viral proteins [20,21]. Hepatitis C virus (HCV)-induced hepatocarcinogenesis is associated with the oncogenic effects of the HCV core antigen and nonstructural protein 5A [NS5A] HCV protein. Unlike HBV, HCV cannot integrate into the host genome [22,23], and it shows a higher propensity of causing chronic infection than HBV, which may result in host immune response evasion and cirrhosis promotion. Conversely, hepatocarcinogenesis induced by aflatoxin B1 is mostly associated with specific tumor protein 53 (*TP53*) mutations [24,25]. Figure 1 shows a diagram showing the development of HCC.

The non-specific mechanisms of hepatocarcinogenesis include alterations resulting from chronic liver diseases, such as the sequence of hepatocytes death, regeneration, and stochastic acquisition of mutations, as well as oncogenic factors derived from the inflammatory environment and the consequences of fibrosis and blood vessel reorganization [1]. For example, alcohol-induced hepatocarcinogenesis is associated with the production of proinflammatory cytokines and the induction of inflammation that result from chronic alcohol intake, leading to consequent cycles of hepatocyte necrosis and regeneration, stellate cell activation, oxidative stress, cirrhosis, and eventually, HCC [20].

## 4. Clinical Features

Patients with HCC may present with clinical signs and symptoms related to the tumor or underlying chronic liver disease [1]. The symptoms include abdominal pain, general malaise, anorexia, weight loss, nausea, and vomiting. Common clinical signs include jaundice, ascites, hepatomegaly, and splenomegaly. An elevated serum α-fetoprotein (AFP) level (>400 ng/mL) strongly supports the diagnosis of HCC. However, most patients with HCC, particularly those in the early stages, do not present with elevated AFP levels. Elevated serum AFP may occur in patients with liver disease without HCC. Thus, AFP levels should be interpreted in the context of clinical and radiological findings. Multifocal HCCs are common in cirrhotic livers and may represent multicentric HCCs that arise simultaneously or as intrahepatic metastases from a primary tumor [26,27]. Intrahepatic spread via the portal veins is the most common route, and its frequency increases with tumor size. HCC invasion into the bile ducts occurs in approximately 5% of cases and lymph node metastases occur in <5% of cases [1].

## 5. Radiological Features

Imaging studies are important for identifying and localizing HCC [1]. Standard imaging techniques include contrast-enhanced ultrasonography, contrast-enhanced computed tomography (CT), and magnetic resonance imaging (MRI) [1]. In patients with HCC, typical findings include enhancement in the arterial phase (wash-in) and hypointensity in the venous phase (wash-out) on contrast-enhanced CT and/or MRI. Accurate identification of early HCCs or small HCCs (<2 cm in diameter) by imaging can be challenging due to the similarities in features between early HCCs and dysplastic nodules [28]. Early HCCs usually appear isovascular or hypovascular, whereas small progressed HCCs appear mostly hypervascular in the arterial phase and hypovascular in the venous phase.

## 6. Molecular Features

The most common mutations found in HCC are found in the *TERT* promoter (60% of tumors), *TP53* (50%), catenin beta 1 (*CTNNB1*) (40%), AT-rich interaction domain 1A (ARID1A) (10–20%), and axis inhibition protein 1 (*AXIN1*) (10–15%). Additionally, fibroblast growth factor 19 (*FGF19*) (5–10% of tumors) and vascular endothelial growth factor A (*VEGFA*) amplifications (5–10%) and DNA methylation in the insulin-like growth factor 2 (*IGF2*) gene were identified in patients with HCC [29,30,31]. Both genetic and epigenetic mechanisms, including *TERT* promoter mutations, chromosomal aberrations, and methylation events, are thought to serve as gatekeepers for the malignant transformation of dysplastic nodules [32]. As HCC progresses from dysplastic nodule to early and advanced HCC, genetic mutations are gradually accumulated [33,34,35]. Table 2 shows common recurrent gene mutations in HCC.

The major deregulated signaling pathways and associated driver genes in HCC are telomere maintenance (*TERT* promoter), cell cycle regulation (*TP53,* ataxia telangiectasia mutated (*ATM*), retinoblastoma 1 (*RB1*), cyclin-dependent kinase 2A (*CDK2A*), myelocytomatosis oncogene (*MYC*), cyclin D1 (*CCND1*)), Wingless-related integration site (Wnt)–β-catenin signaling (*CTNNB1*, *AXIN1,* adenomatous polyposis coli (*APC*)), chromatin remodeling (*ARID1A*, *ARID2*, lysine methyltransferase 2A (*KMT2A*), *KMT2B*, *KMT2C*, BRCA1 associated protein-1 (*BAP1*), *ARID1B*), rat sarcoma (RAS)/phosphoinositide 3-kinase (PI3K)/mammalian target of rapamycin (mTOR) signaling (ribosomal protein S6 kinase A3 (*RPS6KA3*), phosphatidylinositol-4,5-bisphosphate 3-kinase catalytic subunit alpha (PIK3CA), Kirsten rat sarcoma viral oncogene homolog (*KRAS*), neuroblastoma rat sarcoma viral oncogene homolog (*NRAS*), platelet derived growth factor receptor alpha (*PDGFRA*), epidermal growth factor receptor (*EGFR*), phosphatase and tensin homolog (*PTEN*)), fibroblast growth factor signaling (*FGF19*), vascular endothelial growth factor pathway (*VEGFA*), oxidative stress (nuclear factor erythroid 2-related factor 2 (*NFE2L2*), kelch like ECH associated protein 1 (*KEAP1*)), hepatocytic differentiation (albumin (*ALB*), apolipoprotein B (*APOB*)), Janus kinase-signal transducers and activators of transcription (JAK–STAT) signaling (interleukin 6 cytokine family signal transducer (*IL6ST*), Janus kinase 1 (*JAK1*)), transforming growth factor-β (TGF-β) signaling (activin A receptor type 2A (*ACVR2A*)), and insulin-like growth factor (IGF) signaling (IGF 2 receptor (*IGF2R*)) [31].

## 7. Immunohistochemical Features

Immunohistochemistry (IHC) is useful for identifying the lineage of tumor cell differentiation. Markers for hepatocytic differentiation in HCCs include hepatocyte paraffin 1 (Hep Par-1), arginase-1, glypican-3, polyclonal carcinoembryonic antigen (pCEA), CD10, and AFP. In poorly differentiated HCCs, arginase-1 provides better staining than Hep Par-1. pCEA and CD10 are the most specific markers when canalicular staining is observed, but their sensitivity in HCCs is only 45–80%. AFP staining is positive in approximately one-third of HCCs. Bile salt export protein shows canalicular expression and high sensitivity and specificity for HCC [36]. Approximately 20–35% of HCCs show positivity for stem/progenitor cell markers, such as cytokeratin (CK) 19, epithelial cell adhesion molecule (EpCAM), cluster of differentiation 117 (CD117), and cluster of differentiation 133 (CD133) [37]. Table 3 shows IHC, histochemistry markers, and albumin messenger RNA (mRNA) in situ hybridization used for the diagnosis of HCC.

The use of a three-marker panel (enhancer of zeste homolog 2 [EZH2], heat shock protein 70 [HSP70], and glypican-3) can improve the HCC detection rate in liver biopsy tissues [42]. EZH2 is a sensitive HCC marker, but its specificity is very low because almost all the malignant liver tumors examined are positive regardless of their histogenesis [43]. Glutamine synthetase is frequently detected in HCCs. Strong and diffuse glutamine synthetase staining and β-catenin nuclear accumulation support the diagnosis of *CTNNB1*-mutated HCC (CT-HCC). Albumin mRNA in situ hybridization is a sensitive and specific biomarker for HCC [41]. Additionally, intrahepatic cholangiocarcinomas (ICCAs) are also frequently positive for albumin mRNA [44].

## 8. Pathological Features of Conventional Hepatocellular Carcinoma

### 8.1. Macroscopic Features

HCCs vary in color, from yellow to green to tan/brown, depending in part on their fat and bile contents [38]. Necrosis and hemorrhage are frequently present. Most HCCs are soft and bulge out from the cut liver surface. Some HCCs, particularly those in cirrhotic livers, have pseudocapsules composed of fibrotic and inflamed tissues. HCCs have four main macroscopic patterns that are important for clinical staging: (1) a solitary distinct nodule; (2) a large dominant nodule accompanied by multiple smaller satellite nodules, typically within 2 cm of the dominant nodule; (3) numerous small nodules (ranging from tens to hundreds) resembling cirrhotic nodules in the size and shape, known as diffuse or cirrhotomimetic growth patterns; and (4) multiple distinct nodules representing independent primary nodules [1]. Pedunculated HCCs can also occur, which protrude from the liver surface. Rhee et al. [45] found significant differences in the biological behavior of HCC depending on the different gross type and their importance in predicting early recurrence.

### 8.2. Microscopic Features

HCC tumor cells show hepatocytic differentiation, which can be identified by their morphology and/or IHC staining [1]. In conventional HCC, the tumor displays loss of portal tract, thickening of the hepatic plate (>2 cells in thickness), and a reduction or loss of the normal reticulin framework. In addition, HCCs typically exhibit aberrant arterioles in the parenchyma and sinusoidal capillarization. The cytological atypia varies from minimal to marked. Conventional HCCs have three main histological growth patterns: trabecular, pseudoacinar (pseudoglandular), and solid (compact) [1,38]. Approximately 50% of resected HCCs exhibit mixed patterns, primarily trabecular with one or two other patterns.

Some HCCs show bile production, lipofuscin deposits, fatty change, and glycogen accumulation, resulting in clear cell change. The tumor cells may show distinctive inclusions in their cytoplasm, such as hyaline bodies, Mallory–Denk bodies, or pale bodies. Some HCCs have two or more distinct morphologies, with variations in architectural patterns, morphological subtypes, and tumor grades. In most cases, the emergence of a poorly differentiated nodule within an existing HCC indicates tumor progression, a pattern commonly referred to as nodule-in-nodule growth [1].

For tumor grading, two four-tiered grading systems are commonly used: the Edmondson–Steiner and the modified Edmondson–Steiner systems [46,47]. The proposed WHO grading system is a three-tiered grading system that scores tumors as well, moderately, or poorly differentiated [1]. Although the Edmondson–Steiner grading system works reasonably well in research studies, the three-tiered grading system is preferred in clinical practice [38,48]. HCCs may have more than one grade. In such instances, it is essential to report both the worst (even if it is in the minority) and the predominant grades. The worst grade generally determines the prognosis [49].

## 9. Subtypes of Hepatocellular Carcinoma

In addition to the conventional morphological features described, up to 35% of HCCs can be further subclassified into different morphological subtypes, representing distinct clinicopathological and molecular entities [1]. The subtypes of HCC recognized in the 2019 WHO classification are summarized in Table 4.

### 9.1. Steatohepatitic Hepatocellular Carcinoma

Steatohepatitic HCC (SH-HCC) is a morphological subtype of HCC characterized by histological features resembling steatohepatitis, including macrovesicular steatosis, balloon cells, intratumoral inflammation, and intratumoral pericellular fibrosis [92]. The term SH-HCC can be used if these changes are evident in 50% or more of the tumor. The frequency of SH-HCC is 5–20% of all HCC cases [1]. This subtype is most commonly observed in patients with metabolic syndrome [50] or with alcohol use [51], but a small subset of cases lacks these risk factors [52,53,54,93]. Compared to conventional HCC, mutations in *CTNNB1, TP53*, and *TERT* promoter are less frequent; however, the interleukin-6 (IL-6)–JAK–STAT signaling pathway is more frequently activated [93,94].

Histologically, SH-HCC tumors show macrovesicular steatosis, lymphocytic inflammation, balloon cells, Mallory–Denk bodies, and pericellular fibrosis (Figure 2) [55]. Steatohepatitis should be a dominant histological component in at least 50% of the tumor. HCCs that show only mild macrovesicular steatosis do not meet the criteria for the diagnosis of SH-HCC [39]. To be diagnosed as SH-HCC, the tumor should exhibit fat droplets, evidence of injuries such as inflammation or ballooned tumor cells, and pericellular fibrosis, which can be effectively detected using a trichrome stain. Most SH-HCC cases are well to moderately differentiated.

The differential diagnosis of SH-HCC includes steatohepatitis, focal nodular hyperplasia (FNH), and steatotic HCA. In some cases, the tumor can be mistaken for steatohepatitis alone. Correct diagnosis results from the recognition of the architectural and cytological atypia of the tumor cells. The presence of fibrous septa, thick-walled arterioles, ductular reaction, and a map-like pattern of glutamine synthetase-positive hepatocytes favors FNH. The lack of cytologic atypia, lack of thick hepatic cords (>2 cell plates), lack of staining for glypican-3, glutamine synthetase, and HSP70, young age, female gender, and non-cirrhotic background generally favor HCA [95].

### 9.2. Clear Cell Hepatocellular Carcinoma

Clear cell HCC (CC-HCC) has clear tumor cells with clear abundant cytoplasm filled with glycogen [96]. Although clear cells can be seen in many HCCs, according to the 2019 WHO criteria, the tumor must have more than 80% clear cells to be classified as CC-HCC. The frequency of CC-HCC is 3–7% of all HCC cases [1]. *IDH1* mutations have been reported [56]; however, these molecular observations are insufficient to define this subtype. Compared to conventional HCCs, CC-HCC tumors are smaller and better differentiated [57], with lower rates of vascular invasion [57,58,59]. The prognosis of patients with CC-HCC is better than that of conventional HCC [96].

Histologically, the tumor cells have abundant, clear cytoplasm due to glycogen accumulation in the cytoplasm and are typically well or moderately differentiated (Figure 3). In addition to glycogen, some tumor cells can contain fat, and steatosis is present in one-third of CC-HCC cases [60]. CC-HCCs are positive for markers of hepatocytic differentiation, including Hep Par-1, arginase-1, and albumin mRNA in situ hybridization.

The differential diagnosis of CC-HCC includes lipid-rich HCC, angiomyolipoma, and metastatic clear cell neoplasms from other organs, such as the kidney and ovary. Lipid-rich HCCs contain tumor cells filled with tiny lipid droplets but not glycogen. Angiomyolipomas exhibit clear cell changes and are positive for smooth muscle actin and human black melanoma 45 (HMB-45). Immunohistochemical staining is useful for distinguishing metastatic clear cell neoplasms. Clear cell renal cell carcinomas are positive for paired box 8 (PAX8) and negative for Hep Par-1 [61]. Clear cell carcinomas of the ovary can be focally positive for Hep Par-1 [97]. Therefore, it is recommended to use Hep Par-1 in combination with other hepatocytic differentiation markers (e.g., arginase-1) for the differential diagnosis of HCC.

### 9.3. Macrotrabecular Massive Hepatocellular Carcinoma

Macrotrabecular massive HCC (MT-HCC) is an HCC subtype defined by a macrotrabecular growth pattern in more than 50% of the tumor, regardless of the associated cytological features [62,93]. Several cutoff values have been used to define the macrotrabecular growth pattern, but the most common are >6 [98] and ≥10 in thickness [38]; however, some previous definitions also used ≥20 [99]. MT-HCC accounts for 5% of all HCC cases. This subtype is more common in patients infected with HBV. MT-HCCs are usually associated with elevated serum AFP levels, larger tumor size, higher incidence of vascular invasion, and higher grades and stages. MT-HCCs often have *TP53* mutations and/or *FGF19* amplifications [1]. Gene expression profiling has demonstrated that the activation of angiogenesis is a characteristic feature of MT-HCC, with overexpression of both angiopoietin 2 and vascular endothelial growth factor A (VEGFA) [93]. The prognosis of MT-HCC is generally poor [1,63].

Histologically, MT-HCC tumor cells generally have basophilic cytoplasm and show macrotrabecular growth patterns (Figure 4). The tumor cells are often associated with high-grade nuclear atypia, including enlarged, hyperchromatic nuclei and prominent nucleoli. CD34 immunostaining can be used to identify the macrotrabecular growth pattern [100]. Endothelial-specific molecule 1 (ESM1) is expressed by stromal endothelial cells in MT-HCCs and can be a reliable immunohistochemical marker of this subtype [64]. The vessels encapsulating tumor clusters (VETC) pattern, previously associated with metastatic dissemination of HCC, is enriched in MT-HCC [63]. This distinct vascular pattern is a strong pathological finding affecting the aggressiveness of HCC. In biopsy specimens, cases are classified as MT-HCC if at least one focus of the macrotrabecular growth pattern is identified, regardless of the percentage of macrotrabecular growth [101].

The differential diagnosis of MT-HCC includes CC-HCC and macrotrabecular hepatoblastoma [102,103]. CC-HCCs show clear cytoplasm owing to glycogen accumulation and a thin trabecular growth pattern. Macrotrabecular hepatoblastomas are characterized by a thick cell plate (5–12 cells in thickness) and are composed of fetal or embryonal hepatoblasts, pleomorphic cells, or cells resembling those found in HCC [104].

### 9.4. Scirrhous Hepatocellular Carcinoma

Scirrhous HCC is characterized by abundant and diffuse intratumoral fibrosis, which should involve at least 50% of the tumor. It accounts for 4% of all HCCs [1]. Scirrhous HCCs are often located beneath the liver capsule and are more common in non-cirrhotic livers [65,66,67]. The clinical features of scirrhous HCC are similar to those of conventional HCC [96]. Radiological imaging often resembles ICCA [105]. Molecular changes include *TSC1/TSC2* mutations [93] and TGF-β signaling pathway activation [68]. The prognosis varies, with no consensus in the literature [1].

Histologically, scirrhous HCC tumors are well to moderately differentiated in most cases. Fibrosis is diffuse, with more than 50% of the tumor showing dense intratumoral fibrosis [38,68]. Hyaline bodies and pale bodies may be present [69]. Fatty or clear cell changes are also observed [70]. Approximately 50% of scirrhous HCCs are negative for Hep Par-1, whereas approximately 80% of cases are positive for arginase-1 and glypican-3 [70]. The tumor cells frequently express adenocarcinoma-associated markers, such as EpCAM, CK7, and CK19 [70].

The differential diagnosis of scirrhous HCC includes fibrolamellar carcinoma (FLC), ICCA, and metastatic tumors. Both scirrhous HCCs and FLCs are CK7 positive [71], whereas only FLCs consistently express CD68 [72,106]. Molecular testing for the *DNAJB1*::*PRKACA* fusion helps diagnose FLC [107]. Some cases have a scirrhous morphology but display either clear cell change or steatohepatitis morphology, sufficient to qualify for CC-HCC or SH-HCC, respectively [73,105]. Although there is currently no specific guideline for these cases, it is considered reasonable to classify them based on the predominant pattern until further clarification is available [105]. A panel of immunohistochemical markers can be used for the differential diagnosis of ICCAs and metastatic tumors. Positive staining for hepatocytic differentiation markers can rule out ICCAs and metastatic tumors.

### 9.5. Chromophobe Hepatocellular Carcinoma

Chromophobe HCC is a rare subtype of HCC characterized by tumor cells with clear to pale eosinophilic cytoplasm and generally bland nuclear changes; however, scattered tumor cells show prominent nuclear pleomorphism [96]. Chromophobe HCC was first described by Wood et al. [74] in 2013. It accounts for 3% of all HCC cases. Chromophobe HCC is strongly associated with the alternative lengthening of telomeres (ALT), which is a telomerase-independent mechanism that allows telomere length maintenance without *TERT* promoter mutations or other *TERT* gene rearrangements [74,108]. The prognosis of chromophobe HCC is similar to that of conventional HCC [39].

Histologically, the tumor cells have moderately abundant clear to pale eosinophilic cytoplasm and mainly bland nuclei. There are abrupt focal areas of tumor cells with greater nuclear anaplasia [1]. The tumors frequently have scattered cystic spaces filled with a serum-like substance. Chromophobe HCCs are positive for ALT, which can be detected by fluorescent in situ hybridization (FISH) [108,109].

The differential diagnosis of chromophobe HCC includes HCA, CC-HCC, and metastatic chromophobe renal cell carcinoma. HCAs are characterized by benign cells with hepatocellular differentiation with uniform nuclei and low nuclear–cytoplasmic ratio, and fat and glycogen can be abundant in the cytoplasm. CC-HCCs lack the scattered foci of anaplasia, have no cyst formation areas, and are ALT negative. Metastatic chromophobe renal cell carcinomas have large pale cells with prominent cell membranes and perinuclear haloes and are positive for CK7 and CD117.

### 9.6. Fibrolamellar Carcinoma

Fibrolamellar carcinoma (FLC), also known as fibrolamellar HCC, is a rare subtype of HCC composed of neoplastic hepatocytes with prominent intratumoral fibrosis. It accounts for 1% of all HCCs. FLCs commonly occur in young patients aged between 10 and 35 years (mean age, 26 years) [75] with no underlying liver disease. The sex distribution in FLC is about equal. Serum AFP levels can be mildly elevated [76], but never exceed 200 ng/mL. In more than 99% of all FLC cases, the microdeletion of chromosome 19 results in a DnaJ heat shock protein family (Hsp40) member B1 (*DNAJB1*):: protein kinase cAMP-activated catalytic subunit alpha (*PRKACA*) fusion gene. Most cases are sporadic; however, rare cases occur in patients with Carney complex [77]. FLCs associated with the Carney complex have inactivating protein kinase cAMP-dependent type I regulatory subunit alpha (*PRKAR1A*) mutations instead of the *DNAJB1*::*PRKACA* fusion gene. FLCs rarely occur in the same liver as HCAs [78,79,110]. FLCs have more frequent metastases to hilar lymph nodes than conventional HCCs. The overall prognosis is similar to that observed in conventional HCCs without underlying liver disease [80,81].

Histologically, the tumor cells are large and polygonal with abundant eosinophilic cytoplasm and prominent nucleoli with intratumoral dense fibrosis (Figure 5) [53]. Fibrosis can be deposited in parallel or lamellar bands, which are unnecessary for diagnosis. Pseudoglands, pale bodies, and hyaline bodies are commonly seen in FLCs. Intratumoral cholestasis, calcification, copper accumulation, and mucin production can also be observed [111]. Immunohistochemically, the tumor cells are positive for hepatocellular markers, such as Hep Par-1, arginase-1, and albumin mRNA in situ hybridization. A subset of cases is positive for glypican-3 [75]. The co-expression of CK 7 and CD68 is characteristic of FLC [106]. FISH or polymerase chain reaction (PCR) tests can be helpful in identifying the *DNAJB1*::*PRKACA* fusion gene [82].

The differential diagnosis of FLC includes conventional HCC, scirrhous HCC, mixed FLC-HCC, ICCA, and metastatic neuroendocrine tumors. Conventional HCCs lack characteristic cytomorphology, dense lamellar fibrosis, and *DNAJB1*::*PRKACA* fusion gene. Scirrhous HCCs can be morphologically very similar to FLC. Scirrhous HCCs do not exhibit dense lamellar fibrosis, and the *DNAJB1*::*PRKACA* fusion gene is absent in these cases. FLCs with focal pseudoglandular formation and mucin production may be misdiagnosed as ICCA. The tumor cells in ICCAs are negative for hepatocellular markers and lack the *DNAJB1*::*PRKACA* fusion gene. Metastatic neuroendocrine tumors are negative for hepatocellular markers and positive for neuroendocrine markers such as chromogranin and synaptophysin. In most cases, a correct diagnosis of FLC can be made based on a combination of morphological findings and molecular tests.

### 9.7. Neutrophil-Rich Hepatocellular Carcinoma

Neutrophil-rich HCC (NR-HCC), also known as granulocyte colony-stimulating factor (GCSF) producing HCC, is a rare subtype of HCC with abundant intratumoral neutrophils [105]. This subtype was first described by Yamamoto et al. [83] in 1999. It accounts for <1% of all HCCs. NR-HCCs mostly occur in elderly patients with poor prognoses [84,85,86,87]. NR-HCC is characterized by the production of GCSF, a cytokine that plays a critical role in neutrophil production and maturation, leading to extensive intratumoral neutrophil infiltration [96,105]. Because of GCSF production by the tumor, patients have elevated peripheral white blood cell counts, elevated serum IL-6 levels, and often elevated serum C-reactive protein levels [88,112,113].

Histologically, these tumors have numerous tumor-infiltrating neutrophils [105]. The tumor cells are often poorly differentiated. Focal areas of sarcomatoid differentiation are sometimes observed [85,86]. Immunohistochemically, the tumor cells are positive for GCSF [40]. IHC for GCSF can be helpful for confirming the diagnosis but is not widely available [96].

The differential diagnosis of NR-HCC includes conventional ICCA and metastatic carcinoma. ICCAs [112] and metastatic carcinomas [113] can produce GCSF. Therefore, convincing evidence of hepatocytic differentiation is required for NR-HCC diagnosis. The correlation between imaging findings and clinical history is essential for confirming metastatic carcinoma. Treated HCCs may display areas of necrosis and occasionally exhibit focally intense infiltration of neutrophils [39]. These cases should not be classified as NR-HCC [40].

### 9.8. Lymphocyte-Rich Hepatocellular Carcinoma

Lymphocyte-rich HCC (LR-HCC) is a subtype of HCC with abundant intratumoral lymphocytes [105]. The number of lymphocytes should exceed the number of tumor cells in most areas of the neoplasm. It accounts for <1% of all HCCs. The etiology of LR-HCC remains unclear, and it has not been associated with Epstein–Barr virus infection. LR-HCCs have better prognoses than conventional HCCs [1]. Lymphoepithelioma-like HCC (LEL-HCC) is also characterized by dense tumor-infiltrating lymphocytes. The terms LR-HCC and LEL-HCC are often used as synonyms [38]. However, they differ in the histological grade of the HCC. LR-HCC is usually well to moderately differentiated [96]. Conversely, LEL-HCC is a poorly differentiated carcinoma that grows in ill-defined sheets and closely resembles the lymphoepithelioma-like carcinomas of other organs. The relationship between LR-HCC and LEL-HCC remains unclear, but until further knowledge is available, it is reasonable to consider them as separate entities [38]. LEL-HCCs have a higher prevalence in females and predominantly occur in non-cirrhotic livers [89,90]. Focal amplifications of 11q13.3 and 13q34 have been identified in LEL-HCC [91]. Approximately 60% of LEL-HCC cases are microsatellite stable [89].

Histologically, LR-HCC tumors show diffuse and strong lymphocyte infiltration. The number of lymphocytes is greater than the number of tumor cells in most areas (Figure 6) [39]. Most tumor-infiltrating lymphocytes are T lymphocytes, and germinal centers containing B lymphocytes are often present [89,90]. Plasma cell infiltration has also been observed. Expression of programmed death ligand 1 (PD-L1) in neoplastic and inflammatory cells has been reported in LEL-HCCs [114]. Most tumors are well to moderately differentiated. Pale bodies are often observed in the tumor cells. The tumor cells are positive for markers of hepatocytic differentiation, such as Hep Par-1 and arginase-1.

The differential diagnosis of LR-HCC includes conventional HCC, lymphoepithelioma-like ICCA, and metastatic lymphoepithelioma-like carcinomas of other organs, such as the nasopharynx. Some conventional HCCs with mildly diffuse patchy lymphocytosis or dense focal lymphocytic infiltration are incorrectly classified as LR-HCC [39]. Lymphoepithelioma-like ICCAs and metastatic carcinomas can also exhibit a lymphocyte-rich morphology. Therefore, immunohistochemical staining should be performed to demonstrate hepatocytic differentiation [105].

## 10. New Provisional Subtypes of Hepatocellular Carcinoma Not Recognized in the 2019 WHO Classification

Some HCCs are categorized as provisional subtypes due to the limited published data availability [38]. Table 5 shows the clinical, pathological, and molecular features of the new provisional subtypes of HCC that were not included in the 2019 WHO classification.

### 10.1. CTNNB1-Mutated Hepatocellular Carcinoma

*CTNNB1*-mutated HCC (CT-HCC), also known as β-catenin-mutated HCC, are well-differentiated tumors composed of eosinophilic tumor cells growing in thin trabeculae [115,116]. *CTNNB1* encodes β-catenin, a critical intracellular transducer of Wnt signaling, which regulates liver physiology and zonation [140,141]. Mutations in *CTNNB1* lead to the subsequent nuclear accumulation of β-catenin, which interacts with various transcription factors that enhance cell proliferation and survival [140]. Dysregulation of bile salt transporter expression has also been observed in these tumors, which may contribute to their cholestatic phenotype [117]. CT-HCCs are enriched in males and show significant clinical correlations, such as immune cold and a reduced likelihood of responding to immune checkpoint inhibitor therapies [93]. CT-HCC shows characteristic imaging findings, including a high enhancement ratio on gadoxetic acid-enhanced MR imaging and a high diffusion coefficient on diffusion-weighted imaging [118]. The prognosis is generally favorable.

Histologically, the tumor cells are well-differentiated, with a thin trabecular growth pattern and eosinophilic cytoplasm. The tumor is characterized by a pseudoglandular architecture, bile production, and a lack of immune cell infiltration [117,118]. However, these morphological patterns are neither sensitive nor specific for CT-HCC. Approximately 40% of CT-HCCs do not have the classic morphology [93]. CT-HCC tumor cells show nuclear expression of β-catenin and strong diffuse expression of glutamine synthetase.

The differential diagnosis of CT-HCC includes β-catenin (exon 3)–activated HCA and fetal-type hepatoblastoma. β-catenin (exon 3)–activated HCA shows nuclear β-catenin expression and diffuses strong glutamine synthetase expression but lacks cytological atypia and a distinct pseudoglandular architecture [142,143]. Fetal-type hepatoblastomas frequently show nuclear and cytoplasmic expression of β-catenin and overexpression of glutamine synthetase [104]. These hepatoblastomas are common in children without a history of liver disease. They are composed of thin trabeculae of medium-sized cells resembling hepatocytes and often have other non-fetal components [104].

### 10.2. Sarcomatoid Hepatocellular Carcinoma

Sarcomatoid HCC (SHCC) is a rare subtype of HCC composed of prominent malignant spindle cells [96]. It accounts for less than 1% of all HCCs. However, the frequency of sarcomatoid features in autopsy studies is much higher, reaching as high as 10%. This phenomenon is likely due to the enrichment of advanced-stage tumors in autopsy studies [119]. A definite HCC component is required for the diagnosis of SHCC. Furthermore, the spindle cell component should show no evidence of specific mesenchymal differentiation based on morphology or immunostaining [38,39]. SHCC has risk factors similar to conventional HCC [56]. Sarcomatoid changes in HCC occur more frequently after repeated chemotherapy or transarterial chemoembolization [120]. Recently, it has been suggested that the sarcomatoid changes in carcinoma cells might reflect the morphological and biological adaptation to the external environments [144]. Furthermore, cancer can be viewed as an ecological disease involving a multidimensional spatiotemporal ecological and evolutionary process. The prognosis is poor [39].

Histologically, these tumors consist of an HCC and a prominent spindle cell component [38]. Spindle cells, which lack specific mesenchymal differentiation, are often observed as distinct tumor nodules within the tumor, located at the periphery or in the center of the tumor [38,39]. The HCC component is moderately to poorly differentiated [39]. Immunohistochemically, the spindle-shaped tumor cells are vimentin-positive and at least focally keratin-positive but negative or only focally positive for markers of hepatocytic differentiation [38].

The differential diagnosis of SHCC includes carcinosarcoma and metastatic spindle cell tumors, such as metastatic sarcomas and metastatic sarcomatoid carcinomas. Carcinosarcoma is a rare primary liver cancer with both carcinomatous elements (such as HCC or ICCA) and sarcomatous components that exhibit specific mesenchymal differentiation [145]. Conversely, SHCCs show no specific mesenchymal differentiation. If a biopsy specimen shows only spindle cell morphology, metastatic sarcomas and metastatic sarcomatoid carcinomas should be carefully ruled out, even in the cirrhotic liver [38]. A broad panel of hepatocytic differentiation markers is required to identify hepatocytic differentiation and confirm the differential diagnosis.

### 10.3. Lipid-Rich Hepatocellular Carcinoma

Lipid-rich HCC is a rare subtype of HCC characterized by an abundance of intracytoplasmic lipids within the tumor cells. There are limited published data on this type’s clinical and morphological characteristics [121,122]. A previous case report showed that the patient’s liver presented with non-alcoholic fatty liver disease and advanced fibrosis [121]. The prognosis remains unclear [39].

Histologically, the tumor cells have abundant clear cytoplasm filled with numerous tiny fat droplets, similar to microvesicular steatosis [121]. Microvesicular steatosis should be diffuse; however, larger fat droplets are occasionally acceptable. The tumor cells are well to moderately differentiated.

The differential diagnosis of lipid-rich HCC includes CC-HCC, hepatic adrenal rest tumor, metastatic clear cell renal cell carcinoma, and neuroendocrine tumor with clear cell change [123]. Lipid-rich HCCs may be confused with CC-HCC [38]. Clear cell HCCs have abundant clear cytoplasm due to glycogen accumulation, whereas lipid-rich HCCs are instead filled with numerous tiny fat droplets. Hepatic adrenal rest tumors consist of cords of round to polygonal, pale, lipid-rich cells separated by vascular channels or collagen bands and are positive for CD56 and negative for arginase-1 [146]. Metastatic clear cell renal cell carcinomas and neuroendocrine tumors with clear cell change are negative for hepatocytic differentiation markers, such as Hep Par-1 and arginase-1.

### 10.4. Myxoid Hepatocellular Carcinoma

Myxoid HCC is a rare subtype of HCC characterized by abundant myxoid material dissecting through the sinusoids [38]. Myxoid or mucinous extracellular matrix within a primary liver tumor was first described in a case study in 1995 [147]. Primary hepatic tumors showing myxoid changes and no biliary differentiation were characterized in more detail in a series of four HCAs and five HCCs [124]. The cause of the myxoid deposits is unclear. Myxoid HCCs show distinct radiological features, predominantly distinct T2 hyperintensity with thin internal septations and circumscribed, lobulated borders. Additionally, these subtypes display heterogeneous enhancement in the arterial phase, which becomes more homogenous in the delayed phase [125]. The lesions are hypointense on CT and show enhancement features similar to those observed on MRI. The prognosis remains uncertain [39]. To date, no consistent molecular characteristics have been identified [39].

Histologically, these tumors show abundant and distinctive extracellular myxoid material. The tumor sinusoids are expanded by abundant myxoid material, which has shown positive results for Alcian blue staining and may exhibit weak positivity for mucicarmine staining. Immunohistochemically, the tumor cells are positive for Hep Par-1 and arginase-1 and negative for CK 19 and caudal-type homeobox 2 (CDX2) [38]. The tumor cells often show loss of liver fatty acid binding protein expression and diffuse staining of glutamine synthetase [124].

The differential diagnosis of myxoid HCC includes myxoid HCA and ICCA. Myxoid changes have been observed in HCAs [148]. Therefore, tumors should be thoroughly evaluated for HCC [38]. Myxoid HCAs lack cytological atypia, thick cell plates, and reticulin loss. Myxoid HCCs should not be classified as ICCAs. ICCAs show true glandular differentiation and are positive for CK7 and CK19. There is no morphological or immunostaining evidence that suggests the presence of ICCA in myxoid HCCs [124].

### 10.5. Hepatocellular Carcinoma with Syncytial Giant Cells

HCC with syncytial giant cells (HCC-SG) is a rare subtype of HCC characterized by numerous multinucleated giant cells of the syncytial type [96]. Multinucleated syncytial-type giant cells in HCC were first described in a case of HCC in an infant who presented with features of Cushing’s syndrome from bilateral adrenal hyperplasia in 2007 [126]. These multinucleated syncytial-type giant cells are probably hepatocytes in origin based on their distinctive immunophenotype with reactivity for hepatocyte markers such as Hep Par-1 and CK8 [38]. The prognosis of HCC-SG remains unclear, and there are no consistent molecular findings to date [39].

Histologically, the tumor cells have distinctive syncytial-type giant cells, similar to those observed in infantile giant cell hepatitis [126]. These giant cells have numerous nuclei and eosinophilic cytoplasm and show the same nuclear atypia as the other HCC tumor cells. Other tumor areas may show smaller, less differentiated nests of tumor cells on a desmoplastic background [38]. The tumor cells and giant cells are positive for hepatocellular markers (e.g., Hep Par-1) and CK8 and negative for cholangiocytic markers (e.g., CK7 and CK19). An aberrant nuclear β-catenin expression can be seen in the less differentiated areas of the tumor [38].

The differential diagnosis of HCC-SG includes HCC with osteoclast-like giant cells, SHCC with osteoclast-like giant cells, and HCC with anaplastic multinucleated giant cells. HCCs with osteoclast-like giant cells consist of osteoclast-like giant cells with numerous, small, uniform, benign-appearing nuclei and HCC areas [149,150]. SHCCs with osteoclast-like giant cells show spindle-shaped tumor cells and osteoclast-like giant cells [151,152]. Osteoclast-like giant cells are of non-neoplastic histiocytic origin and may be associated with host reactions [151]. HCCs with anaplastic multinucleated giant cells are poorly differentiated and exhibit bizarre and anaplastic multinucleated giant cells. In contrast to HCC with anaplastic multinucleated giant cells, HCC-SGs lack these giant cells.

### 10.6. BAP1 Mutated and Protein Kinase A Activated Hepatocellular Carcinoma

*BAP1* mutated and protein kinase A (PKA) activated HCC (BP-HCC) is a rare subtype of HCC characterized by mutations in *BAP1* and fibrolamellar-like features [127]. *BAP1* is a tumor suppressor gene encoding BRCA1-associated protein-1, which plays a role in DNA repair and cell cycle regulation. Mutations in *BAP1* and PKA activation are thought to play a role in developing this subtype [127]. Patients with BP-HCC are older and have a poorer prognosis than those with FLC. It is more common in female patients without underlying liver disease or cirrhosis. PKA activation and T cell infiltration suggest that these tumors can be treated with PKA inhibitors [127].

Histologically, BP-HCC tumors display at least focally histological features of FLC [105]. They commonly show abundant fibrous stroma, high lymphocytic infiltration, intratumoral steatosis, biliary tract invasion, and perineural invasion [127].

The differential diagnosis of BP-HCC includes FLC. FLCs have abundant cytoplasm, vesicular nuclei, and prominent nucleoli with dense lamellar fibrosis, and harbor the *DNAJB1*::*PRKACA* fusion gene. In a case study of mixed FLC-HCC, the *DNAJB1*::*PRKACA* fusion gene was also detected [153].

### 10.7. Transitional Liver Cell Tumor

Transitional liver cell tumor (TLCT) is a rare subtype of HCC that occurs in older children and adolescents and exhibits an unusual phenotype regarding clinical presentation, histopathology, IHC, and treatment response [128]. Although these tumors have focal areas resembling hepatoblastoma, all histological features are consistent with those of HCC [154]. Clinically, it presents as a large, expanding mass, mostly located in the right liver lobe. Patients typically have a very high level of serum AFP. TLCTs are aggressive, rapidly progressing neoplasms and do not respond to chemotherapy [155]. This tumor subtype has been renamed “hepatocellular malignant neoplasm, not otherwise specified” according to the International Pediatric Liver Tumors Consensus classification [156]. The pathogenesis of TLCT remains unclear, and further studies are needed to fully elucidate its mechanisms.

Histologically, TLCT tumors vary considerably, with a mixture of HCC and hepatoblastoma features [156]. The tumor cells are well to moderately differentiated and often show diffuse growth patterns, lacking a prominent sinusoidal vascular network [155]. There are focal areas of small basophilic, hepatoblastoma-like cells resembling mainly fetal and/or embryonal morphology, and multinucleated giant cells [39,128]. Immunohistochemically, the tumor cells are positive for AFP. Acinar structures staining for CK7 and CK19 may also be present. The tumor cells express β-catenin in a mixed nuclear, cytoplasmic, or membranous pattern [155].

The differential diagnosis of TLCT includes hepatoblastoma and HCC. TLCTs are often mistaken for hepatoblastomas and conventional HCC. Hepatoblastomas frequently show a biphasic pattern with fetal and embryonal cells. Most of the cases that have both conventional HCC and hepatoblastoma features can be reasonably classified as conventional HCCs, especially those that arise in teenage patients [154].

### 10.8. Cirrhotomimetic Hepatocellular Carcinoma

Cirrhotomimetic HCC, also known as diffuse HCC, is a rare subtype of HCC characterized by numerous small HCC nodules. The number of HCC nodules is usually >30, which are the same size as the cirrhotic nodules in the background liver [39]. The precise biological mechanisms underlying this unusual growth pattern remain unclear. Portal vein tumor thrombosis is present in almost cases [129]. Autopsy studies suggest that the tumor is often present in the large hilar portal veins, seeding the rest of the liver [130]. The prognosis of cirrhotomimetic HCC is poor [130].

Histologically, the tumor cells are well or moderately differentiated, with predominantly trabecular and pseudoglandular growth patterns [131]. Perinodular sclerotic rims, cholestasis, Mallory bodies, and small vessel invasion are often present [132]. The tumor nodules blend with the background cirrhotic liver; furthermore, the nodules may coalesce to form a grossly evident mass in the background cirrhotic liver [39]. Immunohistochemically, the tumor cells are positive for Hep Par-1 and glypican-3 and negative for AFP, CK7, and CK19 [132].

The differential diagnosis of cirrhotomimetic HCC includes HCC with satellite nodules. Conventional HCCs often have satellite nodules, with the number of satellite nodules typically being less than 10 [40]. Additionally, the satellite nodules are usually 1–2 cm in size and close to the main tumor mass, whereas tumor nodules in cirrhotomimetic HCCs are more widely dispersed [39].

### 10.9. Progenitor Hepatocellular Carcinoma

Progenitor HCC, also known as CK19-positive HCC or HCC with CK19 expression, is defined by the expression of stemness-related markers, such as CK19, in more than 5% of tumor cells [133,134,135]. This tumor cell phenotype may result from the dedifferentiation of neoplastic hepatocytes or reflect the malignant transformation of hepatic stem/progenitor cells [135]. A progenitor cell phenotype can be acquired after transarterial chemoembolization and radiofrequency ablation [136,137,157,158]. Progenitor HCC accounts for <5% of all HCCs. This tumor is associated with *TP53* mutations and chromosomal instability [135,138]. Progenitor HCC is characterized by increased serum AFP levels, frequent vascular invasion, increased infiltrative growth, large tumors, poor histological differentiation, multiplicity, higher recurrence rates, and higher rates of resistance to locoregional therapy [138]. The prognosis of this tumor subtype is poor [159].

Histologically, progenitor HCCs are compatible with conventional HCC. Intratumoral fibrous stroma is commonly present [138]. Immunohistochemically, tumor cells are positive for stem/progenitor cell markers, such as CK 19, EpCAM, cluster of differentiation 56 (CD56), CD117, and spalt-like transcription factor 4 (SALL4). CK19 expression in HCC is regulated by fibroblast-derived hepatocyte growth factor via the mesenchymal–epithelial transition (MET)–extracellular signal-regulated kinase 1/2 (ERK1/2)–activator protein 1 (AP1) and specificity protein 1 (SP1) axis and is closely associated with α-smooth muscle actin-positive cancer-associated fibroblasts [159]. The MET signature is commonly observed in CK19-positive HCCs, supporting the invasive biological properties of these HCCs [138].

The differential diagnosis of progenitor HCC includes HCC with stem/progenitor cell features/phenotypes, ICCA, and combined hepatocellular-cholangiocarcinoma (HCC-CCA). According to consensus terminology for primary liver carcinomas, HCCs with stem/progenitor cell features/phenotypes are defined as HCCs composed of small cells with scant cytoplasm, a high nuclear–cytoplasmic ratio, and hyperchromatic nuclei. They are highlighted by stem/progenitor cell markers, such as CK19, CD56, EpCAM, CD117, and others [160]. These cells are often found at the interface between a nest of carcinoma and the adjacent fibrous or desmoplastic stroma [139]. Progenitor HCCs are morphologically compatible with conventional HCCs but do not possess the small cell morphology features observed in HCCs with stem/progenitor cell features/phenotypes. ICCAs lack the hepatocellular differentiation observed in progenitor HCC. Combined HCC-CCAs show features of both HCC and ICCA.

## 11. Molecular and Immune Classes

### 11.1. Molecular Classes

Multiple studies using genomic, epigenomic, immunological, and clinicopathological analyses have identified distinct molecular categories of HCC (Figure 7) [2,161,162,163,164,165,166,167]. The molecular classes of HCC are determined by the major molecular drivers and signaling pathways or by the immune status of the tumor [162,163,165,166,167]. HCC can be divided into two major molecular classes: the proliferation class and the non-proliferation class [135,164,165,166,167]. These molecular classes are closely associated with genomic abnormalities, pathological features, and clinical outcomes.

The proliferation class accounts for approximately 50% of HCC cases. It is characterized by more aggressive tumors with poor histological differentiation, a common association with HBV-related HCC, high vascular invasion, and elevated AFP levels [133]. Furthermore, this class is associated with increased chromosomal instability, frequent TP53 mutations, and *FGF19* and *CCND1* amplifications [164]. This class can be further subdivided into two subclasses: the proliferation–progenitor cell subclass and the proliferation–Wnt–TGF-β subclass. The proliferation–progenitor cell subclass is defined by the transcriptional and protein overexpression of hepatic progenitor markers and corresponds to S2 or iCluster 1 of the Cancer Genome Atlas (TCGA) [165,166]. This subclass accounts for approximately 30% of HCCs [164,165]. It shows activation of prosurvival and cell proliferation signaling pathways such as cell cycle, mTOR, RAS–mitogen-activated protein kinase, MET, insulin-like growth factor-1 receptor, and Ak strain transforming [162]. The inactivating mutations in *AXIN1* and *RPS6KA3* are demonstrated in this subclass. It also expresses progenitor cell markers, such as CK 19 and EpCAM. The proliferation–WNT–TGF-β subclass corresponds to S1 or iCluster 3. This subclass accounts for approximately 20% of HCCs and is characterized by the activation of the Wnt and TGF-β signaling pathways.

The non-proliferation tumor class accounts for approximately 50% of HCCs and corresponds to iCluster 2 of the TCGA [135,164,165,166,167]. This class is characterized by less aggressive tumors, well to moderate histological differentiation, less frequent vascular invasion, and low levels of AFP [133,135,164,165,166,167]. The non-proliferation class is associated with non-alcoholic steatohepatitis, alcoholic steatohepatitis, and HCV infection. The non-proliferation class can be subdivided into two specific subclasses: the Wnt–β-catenin *CTNNB1* subclass and the interferon subclass. The Wnt–β-catenin *CTNNB1* subclass presents frequent *CTNNB1* mutations and activates the Wnt–β-catenin signaling pathway [164,167,168]. *TERT* promoter mutations are common in this subclass. The interferon subclass presents a highly activated IL-6–JAK–STAT signaling pathway [164].

### 11.2. Immune Classes

Several studies have attempted to establish a classification of HCC based on immune characteristics [2,169]. Based on the clustering of immune-related gene-expression signatures, an extensive immunogenomic analysis identified six immune subtypes (C1–C6), among which the inflammatory (C3) subtype and the lymphocyte-depleted (C4) subtypes are dominant in HCCs [170]. Sia et al. [171] found that approximately 25% of HCCs exhibit markers of the inflammatory response, characterized by high expression levels of PD-L1 and programmed cell death protein 1 (PD-1), markers of cytolytic activity, and fewer chromosomal aberrations. They called this tumor subset the “Immune class”.

HCCs can be divided into two major immune classes (Figure 8): the inflamed class (~35% of HCCs) and the non-inflamed class (~65% of HCCs) [169,172]. The inflamed class includes three subclasses: the immune active subclass, the immune exhausted subclass, and the immune-like subclass. The immune active subclass is characterized by high levels of CD8+ T cells and M1 macrophage infiltration, enrichment of interferon signaling, overexpression of genes related to the adaptive immune response, and a favorable prognosis. In contrast, the immune-exhausted subclass is characterized by an abundance of M2 macrophages and high levels of TGF-β signaling, T cell exhaustion, stromal activation, and immunosuppressive components. The immune-like subclass has similar features to the immune-active subclass but has increased Wnt–β-catenin signaling. Recently, the inflamed class was shown to be more common in patients with HCC who respond to anti-PD-1/PD-L1 antibodies [173]. The non-inflamed class includes two distinct subclasses: the immune intermediate and the immune excluded subclass. The immune intermediate subclass is linked with a high frequency of *TP53* mutations, a high degree of chromosomal instability, and frequent losses of genes related to antigen presentation and interferon signaling. The immune excluded subclass is characterized by high Wnt–β-catenin signaling levels through *CTNNB1* mutations and immune desertification features.

HCC can be divided into four different immunovascular subtypes: immune-high/angiostatic (IH/AS), immune-mid/angio-mid (IM/AM), immune-low/angiogenic (IL/AG), and immune-low/angio-low (IL/AL) [174]. This classification represents the reciprocal interaction between immune and angiogenic tumor microenvironments and can provide valuable insights into the therapeutic effectiveness of immunotherapy, antiangiogenic therapy, and their combinations. Tertiary lymphoid structures (TLSs) are defined as lymphoid aggregates that form in non-hematopoietic organs in response to chronic and non-resolved inflammatory processes [175]. TLSs provide a crucial local microenvironment for the antitumor immune response. According to the study by Calderaro et al. [176], intra-tumoral TLSs were associated with a decreased risk of early HCC recurrence after surgery. They suggested that TLSs might be an indication of effective antitumor immune responses.

## 12. Pathological Diagnostic Approach

### 12.1. Specimen Handling

#### 12.1.1. Biopsy Specimens

The pathological diagnosis of HCC is relatively difficult when using needle biopsy specimens; thus, biopsy analysis can be aided by careful imaging evaluations [177]. Biopsies should be performed on the different components of radiologically heterogeneous masses. A complete biopsy sample should be measured and submitted for routine histology unless special tests are necessary [178]. Hematoxylin and eosin (H&E) staining is the standard staining method used for liver pathology [179]. IHC plays a crucial role in the evaluation of liver tumors. Step sections are preferred over serial sections because the intervening sections are then available for histochemical and IHC staining. Fresh tumor samples can be used for molecular analysis.

#### 12.1.2. Hepatectomy Specimens

There are no clear guidelines for the pathological assessment of the hepatectomy specimens. It is recommended to cut the liver serially, perpendicular to the resection margin, at thin intervals (e.g., 0.5 cm) and carefully examine all cut surfaces for tumor nodules [178]. When grossly examining the liver mass lesions, pathologists should assess the number and size of tumor nodules, the presence or absence of tumor rupture, macroscopic vascular invasion, and the distance of the tumor from the inked surgical margin, and evaluate any macroscopic changes in the non-neoplastic liver, such as cirrhosis, steatosis, etc. In each case, the tumor size and the presence or absence of tumor rupture should be documented [1]. Additionally, pathologists should also record findings such as tumor color, consistency, cystic and degenerative changes, and necrosis in the gross description. Particularly for tumors up to 2 cm, the entire tumor should be examined microscopically. For larger tumors, at least one block per 1 cm of tumor is recommended. All different tumor areas and transition areas between other areas should be sampled [180,181].

The resection margin should be carefully examined because a positive margin, either grossly or microscopically, carries a higher risk of recurrence [182]. In a previous study for macroscopically solitary HCCs, especially HCC ≤ 2 cm, a wide resection margin of 2 cm provided better survival outcomes than a narrow resection margin of 1 cm [183]. Microscopic vascular invasion is a well-known risk factor for disease recurrence. The interface between tumor and non-tumor tissues is the most suitable site for identifying vascular invasion. When evaluating resected specimens, it is necessary to include sections from the background liver to assess for pre-existing liver disease and fibrosis [38]. It is advisable to obtain background sections of the liver at least 1 cm from the tumor to avoid potential secondary inflammation and fibrosis caused by the tumor’s proximity.

### 12.2. Pathological Diagnostic Approach

HCC diagnosis requires a representative biopsy of the lesion and well-processed tissue. A systemic diagnostic approach leads to an accurate diagnosis of liver mass lesions [184]. Careful histological evaluation of H&E-stained sections at low magnification is the first and most crucial step in the diagnostic approach. Pathologists should initially determine whether the lesional tissue is present and examine the morphology of tumor cells, the growth (architectural) patterns, and the characteristics of the stroma. Clinical information, radiological findings, and serum tumor markers (e.g., AFP and carbohydrate antigen 19-9) can aid in the diagnosis of HCC. IHC and molecular analyses are helpful in diagnosing difficult cases and identifying unusual subtypes of HCCs [185]. Figure 9 shows the diagnostic algorithm used for HCC.

## 13. Non-Invasive Biomarkers

In recent years, non-invasive biomarkers such as circulating tumor DNAs (ctDNAs) and extracellular vesicles (EVs) have been widely used in oncology research [186]. ctDNA are DNA fragments that originate from necrotic or apoptotic cells and can be detected in peripheral blood. ctDNA, characterized by DNA aberrations and evaluated by DNA integrity (the ratio of ctDNA to the whole circulating DNA length), can serve as a biomarker for early diagnosis, therapeutic monitoring, and prediction of tumor recurrence in HCC [186,187,188]. 

EVs are lipid bilayer-enclosed particles released by almost all cell types and can be categorized into exosomes and microvesicles based on their size. They play a role in intercellular communication. For example, EVs mediate interactions between tumor cells, immune cells, and other components of the tumor microenvironment in HCC [189]. Tumor-derived exosomes may be involved in resistance to the chemotherapeutic agents in HCC [190]. Consequently, EVs can be potential biomarkers and therapeutic targets in HCC.

MicroRNAs (miRNAs) are a class of small non-coding RNAs, usually 19–25 nucleotides in length, that play a crucial role in regulating gene expression within cells. For example, circulatory miR-542 is significantly upregulated in HCV-associated cirrhosis and chronic HCV groups but is markedly downregulated in HCV-associated HCC groups [191]. In contrast, circulatory miR-21 and miR-122 are significantly upregulated in HCC groups compared to healthy and cirrhotic groups [192]. Circulatory miRNAs are increasingly recognized as potential non-invasive biomarkers for HCC.

## 14. Future Perspectives

Our understanding of HCC biology has significantly improved over the last two decades. However, a well-defined, easy-to-use, reproducible, and broadly adopted HCC grading system still needs to be developed. Additionally, it is necessary to establish diagnostic criteria for provisional subtypes of HCC in the future. Further integrative pathological, molecular, and immunological classification is needed to understand the biology of HCC. Recently, Murai et al. [193] identified the link between intratumoral steatosis and immune-exhausted immunotherapy-susceptible tumor immune microenvironment through genomic and transcriptomic analyses. Novel biomarkers with greater specificity are required to improve the early detection, treatment, and prognosis of HCC [194,195,196]. Despite the numerous mutations identified in HCC, developing targeted therapies remain challenging. A better understanding of intra- and inter-tumor heterogeneity is needed to understand their role in primary and secondary resistance to systemic treatments [163]. Some molecular techniques, such as next-generation sequencing and microarray platforms, are useful for detecting alterations in nucleic acids, including DNA and RNA [197]. These advanced technologies will significantly contribute to our understanding of the biology of HCC, refining its classification, improving prognostication, and guiding therapeutic strategies.

### 14.1. Treatment of Hepatocellular Carcinoma

HCC is an aggressive tumor. While surgical treatment can offer a curative option for early and resectable HCCs, patients with inoperable advanced HCCs often rely on alternative therapeutic approaches. Current and promising treatments include systemic therapies with multikinase inhibitors such as sorafenib and lenvatinib, molecularly targeted therapies, and immunotherapies, notably immune checkpoint inhibitors [198]. The rapidly evolving therapeutic landscape of advanced HCC warrants further research [199].

### 14.2. Artificial Intelligence

Artificial intelligence (AI) represents a form of machine intelligence, encompassing computational search algorithms, machine learning, and deep learning models [200]. Deep learning, for instance, has demonstrated its capability to predict RNA-Seq expression of tumors from whole-slide images [201]. Such models are increasingly utilized to classify cancer types and identify gene mutations [202]. AI combined with digital pathology is poised to become an effective tool for HCC diagnosis, classification, risk assessment, treatment planning, and prediction of treatment outcomes [200,201]. However, further research is still needed to standardize AI algorithms and evaluate them rigorously through prospective studies. This will ultimately enhance their interpretability, generalizability, and transparency [200,203].

## 15. Conclusions

HCC is the most common primary malignant liver tumor with hepatocellular differentiation and shows very heterogeneous features at the molecular and morphological levels. Recent advancements in the histological and molecular classification of HCC have significantly improved our understanding of the biology of the disease. Further studies on the molecular and genetic characterization of HCC are needed to understand its biology better and to identify new biomarkers and potential therapeutic targets.

## Figures and Tables

**Figure 1 biomedicines-11-02582-f001:**
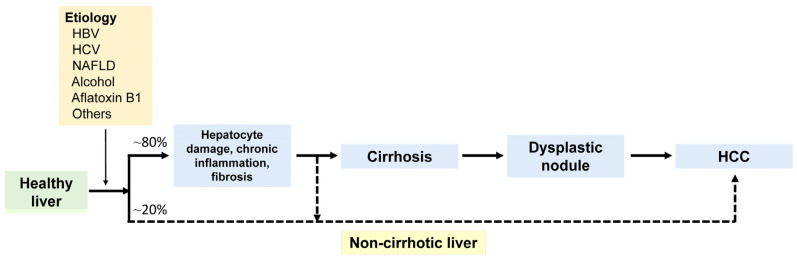
The diagram illustrates the development of hepatocellular carcinoma (HCC) from a healthy liver. Major etiological factors include HBV, HCV, NAFLD, and alcohol. HCC can arise within a dysplastic nodule in cirrhosis through a multistep process or develops de novo in a non-cirrhotic liver. HBV, hepatitis B virus; HCV, hepatitis C virus; NAFLD, non-alcoholic fatty liver disease.

**Figure 2 biomedicines-11-02582-f002:**
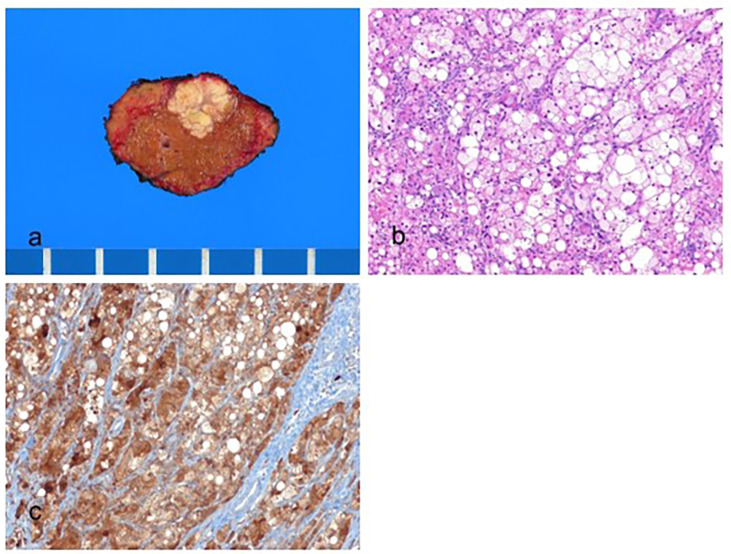
Steatohepatitic hepatocellular carcinoma. (**a**) A multinodular confluent, yellow–gray mass is present. (**b**) The tumor cells show fat droplets and ballooning degeneration (hematoxylin–eosin stain, ×100). (**c**) The tumor cells are positive for hepatocyte paraffin-1 (Hep Par-1) (immunohistochemical stain for Hep Par-1, ×100).

**Figure 3 biomedicines-11-02582-f003:**
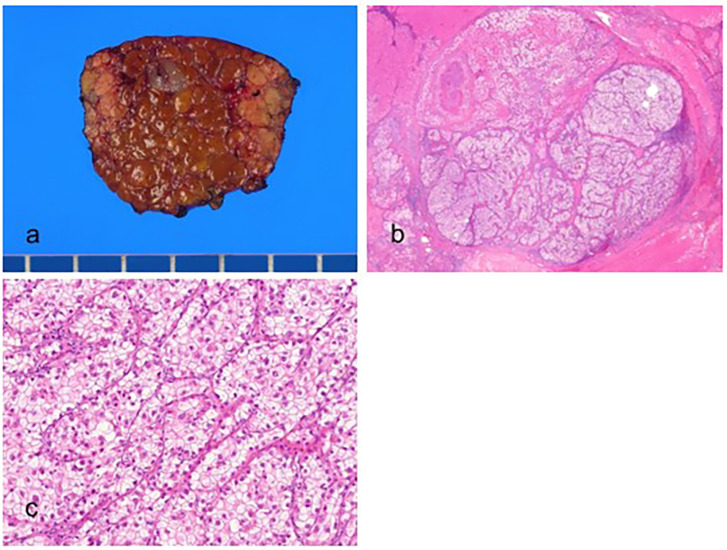
Clear cell hepatocellular carcinoma. (**a**) The vaguely nodular, grayish tumor is present in the cirrhotic background liver. (**b**) Nests of tumor cells are present (hematoxylin–eosin (H&E) stain, ×10). (**c**) The tumor cells have abundant clear cytoplasm (H&E stain, ×100).

**Figure 4 biomedicines-11-02582-f004:**
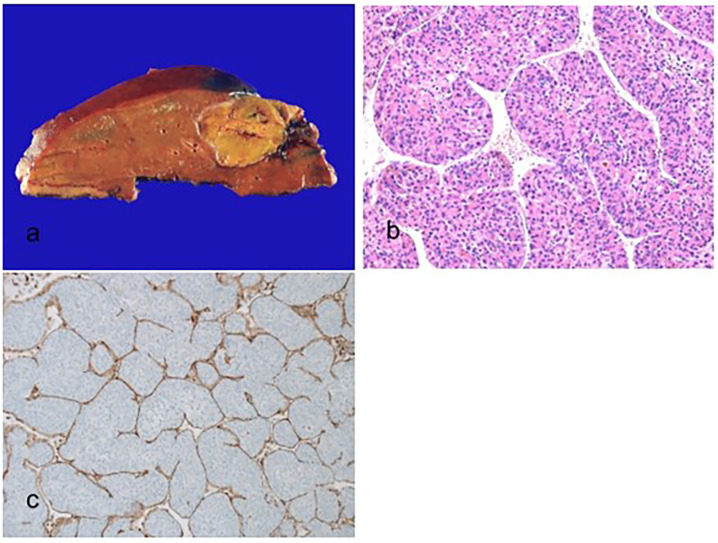
Macrotrabecular massive hepatocellular carcinoma. (**a**) A nodular expanding brown–tan tumor is present. (**b**) The thick trabeculae of more than 10 cells are surrounded by vascular spaces (hematoxylin–eosin stain, ×100). (**c**) The vessels that encapsulate tumor clusters pattern, which encapsulates and separates individual tumor clusters, are present (immunohistochemical stain for CD34, ×100).

**Figure 5 biomedicines-11-02582-f005:**
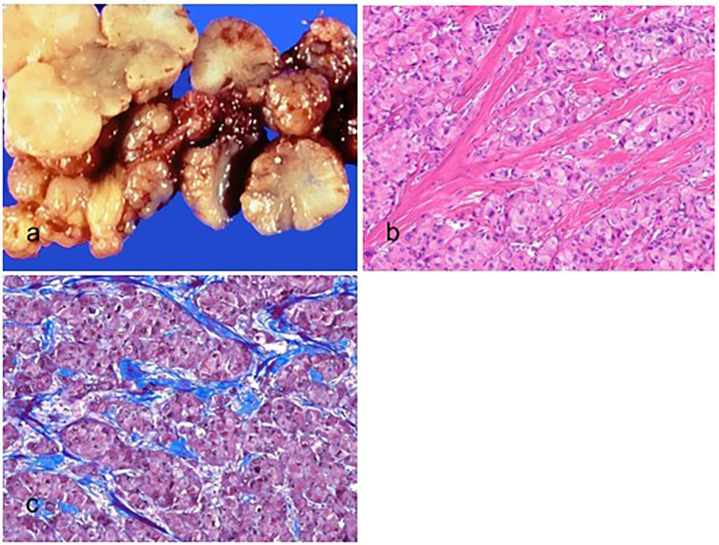
Fibrolamellar carcinoma. (**a**) The tumor is multilobulated, well-circumscribed, yellow, and firm with a central scarring area. (**b**) The tumor cells are large, polygonal, and have abundant eosinophilic cytoplasm and prominent nucleoli (hematoxylin–eosin stain, ×100). (**c**) The intratumoral fibrosis is observed in parallel bands (Masson trichrome stain, ×100).

**Figure 6 biomedicines-11-02582-f006:**
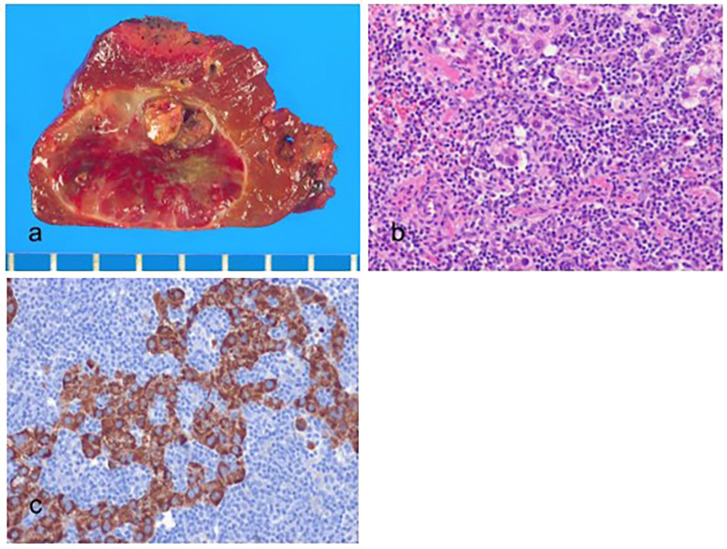
Lymphocyte-rich hepatocellular carcinoma. (**a**) The tumor is well-circumscribed, reddish brown to tan. (**b**) Scattered tumor cells and numerous tumor-infiltrating lymphocytes are present. The tumor-infiltrating lymphocytes outnumber the tumor cells (hematoxylin–eosin stain, ×100). (**c**) The tumor cells are positive for hepatocyte paraffin-1 (Hep Par-1) (immunohistochemical stain for Hep Par-1, ×100).

**Figure 7 biomedicines-11-02582-f007:**
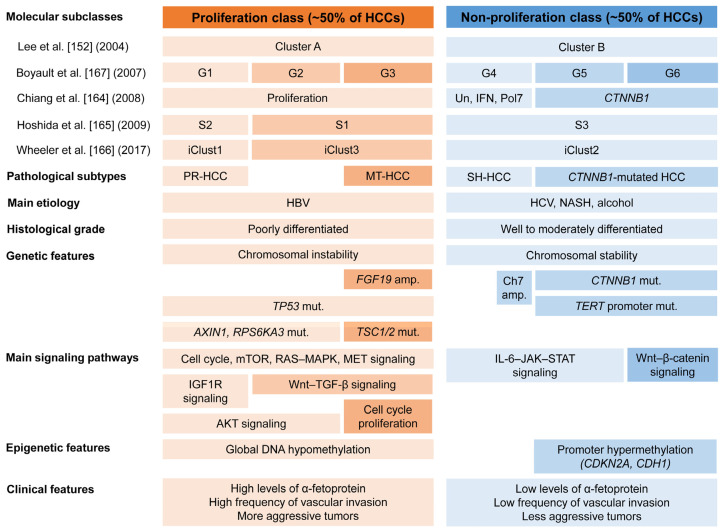
Molecular classes and subclasses of hepatocellular carcinoma. HCCs are classified into two major molecular classes: the proliferation and non-proliferation class. For each molecular subclass, pathological subtypes, main etiology, histological grade, main signaling pathway, genetic, epigenetic, and clinical features are displayed. Macrotrabecular massive HCC is assigned to the G3 subgroup. Progenitor HCC is associated with the S2 or iClust1 subgroup. A subset of HCC within the G4 subgroup frequently exhibits ST-HCC subtype. HCC, hepatocellular carcinoma; Un, unannotated class; IFN, interferon-related class; Pol7, class defined by polysomy of chromosome 7; iClust, integrated cluster; PR-HCC, progenitor HCC; MT-HCC, macrotrabecular massive HCC; SH-HCC, steatohepatitic HCC; HBV, hepatitis B virus; HCV, hepatitis C virus; NASH, non-alcoholic steatohepatitis; amp., amplification; Ch7, chromosome 7; mut., mutation; mTOR, mammalian target of rapamycin; MAPK, mitogen-activated protein kinase; MET, mesenchymal–epithelial transition; IGF1R, insulin-like growth factor 1 receptor; AKT, Ak strain transforming; TGF-β, transforming growth factor-β; IL-6, interleukin-6; JAK, Janus kinase; STAT, signal transducer and activator of transcription [135,164,165,166,167].

**Figure 8 biomedicines-11-02582-f008:**
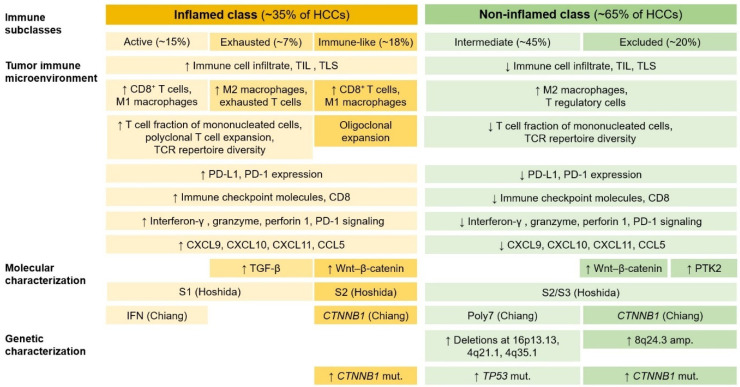
Immune classes and subclasses of hepatocellular carcinoma. For each immune subclass, tumor immune microenvironment and molecular and genetic characterization are shown. HCCs can be divided into two major immune classes: the inflamed class and non-inflamed class. The inflamed class includes three distinct subclasses: the immune active, immune exhausted, and immune-like subclass. The non-inflamed class is composed of the immune intermediate and immune excluded subclasses. HCC, hepatocellular carcinoma; PD-L1, programmed death-ligand 1; PD-1, programmed cell death protein 1; TIL, tumor-infiltrating lymphocyte; TLS, tertiary lymphoid structure; TCR, T-cell receptor; CXCL, C-X-C motif chemokine ligand; CCL5, C-C motif chemokine ligand 5; TGF-β, transforming growth factor-β; PTK2, protein tyrosine kinase 2; IFN, interferon-related class; Pol7, class defined by polysomy of chromosome 7; amp., amplification; mut., mutation; ↑, increase; ↓, decrease.

**Figure 9 biomedicines-11-02582-f009:**
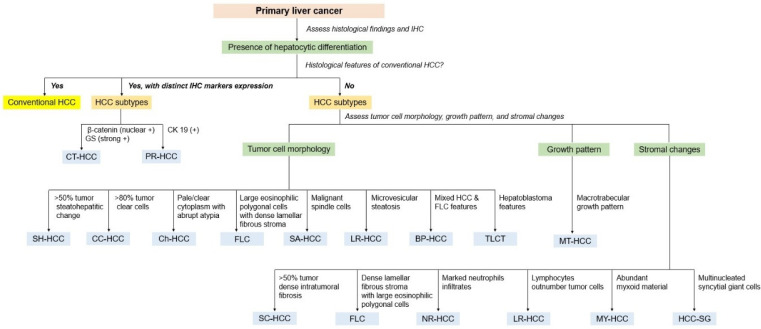
Pathological diagnostic algorithm of subtypes of hepatocellular carcinoma according to tumor cell morphology, growth pattern, stromal change, and immunohistochemistry. IHC, immunohistochemistry; HCC, hepatocellular carcinoma; GS, glutamine synthetase; CK 19, cytokeratin 19; CT-HCC; *CTNNB1*–mutated hepatocellular carcinoma, PR-HCC, progenitor hepatocellular carcinoma; SH-HCC, steatohepatitic hepatocellular carcinoma; CC-HCC, clear cell hepatocellular carcinoma; Ch-HCC, chromophobe hepatocellular carcinoma; FLC, fibrolamellar carcinoma; SA-HCC, sarcomatoid hepatocellular carcinoma; LR-HCC, lipid-rich hepatocellular carcinoma; BP-HCC, *BAP1* mutated and PKA activated hepatocellular carcinoma; TLCT, transitional liver cell tumor; MT-HCC, macrotrabecular massive hepatocellular carcinoma; CM-HCC, cirrhotomimetic hepatocellular carcinoma; SC-HCC, scirrhous hepatocellular carcinoma; NR-HCC, neutrophil-rich hepatocellular carcinoma; LR-HCC, lymphocyte-rich hepatocellular carcinoma; MY-HCCC, myxoid hepatocellular carcinoma; HCC-SG, hepatocellular carcinoma with syncytial giant cells.

**Table 1 biomedicines-11-02582-t001:** Evolution of the World Health Organization (WHO) classification of hepatocellular carcinoma.

	2000 WHO Classification (3rd Edition)	2010 WHO Classification (4th Edition)	2019 WHO Classification (5th Edition)
Tumor category	Epithelial tumors, malignant	Epithelial tumors: hepatocellular, malignant	Malignant hepatocellular tumors
Tumor subtypes	HCC	HCC	HCC, NOS
	Fibrolamellar carcinoma	Fibrolamellar carcinoma	Fibrolamellar carcinoma
		Scirrhous HCC	HCC, scirrhous
		Lymphoepithelioma-like carcinoma	HCC, clear cell
		Sarcomatoid HCC	HCC, steatohepatitic
			HCC, macrotrabecular massive
			HCC, chromophobe
			HCC, neutrophil-rich
			HCC, lymphocyte-rich

HCC, hepatocellular carcinoma; NOS, not otherwise specified.

**Table 2 biomedicines-11-02582-t002:** Common recurrent gene mutations in hepatocellular carcinoma ^(a)^.

Genes	Approximate Frequency	Signaling Pathways and Roles
*TERT* promoter	60%	Controls the transcription of the *TERT* gene which encodes for the catalytic subunit of telomerase; increased TERT expression and telomerase activity in HCC cells
*TP53*	50%	DNA repair and surveillance; frequency varies by risk factors, with the highest risk in areas of chronic aflatoxin B1 exposure
*CTNNB1*	40%	Wnt–β-catenin signaling pathway
*ARID1A*	10–20%	Chromatin remodeling
*AXIN1*	10–15%	Wnt–β-catenin signaling pathway
*FGF19* amplification	5–10%	Encodes FGF19 protein that is a member of the FGF family; regulates bile acid synthesis and hepatocyte proliferation by activation of its receptor FGFR4
*RB1*	5–10%	DNA repair and surveillance
*RPS6KA3*	5–10%	Oncogenic MAPK signaling
*NFE2L2*	5%	Oxidative stress
*TSC1/2*	5%	Tumor suppressor genes that encode TSC1 (hamartin) and TSC2 (tuberin), respectively; controls mTOR signaling by inhibiting the activity of the mTOR complex 1
*CDKN2A*	5%	DNA repair and surveillance
*CCND1*	5%	Encodes cyclin D1 protein which plays a critical role in cell cycle progression

^(a)^ Data are based on the 2019 WHO classification (5th edition) of liver tumors [1]. *TERT*, telomerase reverse transcriptase; HCC, hepatocellular carcinoma; FGF, fibroblast growth factor; FGFR4, fibroblast growth factor receptor 4; MAPK, mitogen-activated protein kinase; TSC, tuberous sclerosis complex; mTOR, mammalian target of rapamycin.

**Table 3 biomedicines-11-02582-t003:** Immunohistochemical markers, albumin mRNA in situ hybridization, and histochemistry for diagnosis of hepatocellular carcinoma.

Markers	Nature	Staining Pattern	Approximate Sensitivity ^(a)^	Notes	References
* **Markers for Hepatocytic Differentiation** *					
Arginase-1	Manganese metalloenzyme; plays a crucial role in the urea cycle; expressed in hepatocytes	Cytoplasmic and nuclear	45–95%	Better than Hep Par-1 in poorly differentiated HCCs; negative in 10% of well-differentiated HCCs	[1,38,39,40]
Hep Par-1	Carbamoyl phosphate synthetase 1, rate-limiting enzyme in the urea cycle; located in hepatocyte mitochondria	Cytoplasmic	70–85%	Better than arginase-1 in well-differentiated HCCs	[1,38,39,40]
Polyclonal CEA	Glycoprotein produced during fetal development; cross-react with biliary glycoprotein I on the surface of bile canaliculi	Canalicular	45–80%	Limited sensitivity in poorly differentiated HCCs; ambiguous staining patterns are not uncommon in moderately to poorly differentiated HCCs	[1,38,39,40]
CD10	Zinc-dependent metalloproteinase; expressed along the bile canaliculi and luminal borders of bile ducts	Canalicular	50–75%	Limited sensitivity in poorly differentiated HCCs; ambiguous staining patterns are not uncommon in moderately to poorly differentiated HCCs	[1,38,39,40]
α-fetoprotein	Oncofetal protein produced by the fetal liver and yolk sac during fetal development	Cytoplasmic	30%	Frequently negative in well-differentiated HCCs	[1,38,39,40]
Bile salt export protein	Transmembrane protein; plays a critical role in the transport of bile salts; exclusively expressed at the bile canaliculi	Canalicular	90%	High sensitivity and specificity for HCCs	[36]
Albumin mRNA in situ hybridization	Protein synthesized only in liver	Cytoplasmic	>95%	Positive in 80–95% of cholangiocarcinomas and some cases of metastatic adenocarcinomas	[38,39,40,41]
* **Markers for distinguishing benign and malignant hepatocellular tumors** *					
Glypican-3	Heparin sulfate proteoglycan; play a role in the regulation of cell differentiation and growth	Cytoplasmic	50–80%	More likely to be positive in poorly differentiated HCCs than Hep Par-1 and arginase-1; not expressed in normal and livers and therefore a good marker for malignant liver lesions	[1,38,39,40]
Glutamine synthetase	Enzyme involved in nitrogen metabolism; expressed in normal hepatocytes around the central vein	Cytoplasmic	80%	Strong and diffuse staining supports the diagnosis of HCC in cirrhotic liver; also positive for focal nodular hyperplasia (map-like pattern) and hepatocellular adenoma	[1,38,39,40].
Heat shock protein 70	Highly conserved protein; plays a critical role in maintaining the function and survival of liver cells under stress	Cytoplasm	50–60%	Could be a sensitive marker for the differential diagnosis of early HCC from precancerous lesion or noncancerous liver; low expression in normal hepatocytes and bile ducts	[1,38]
β-catenin	Protein encoded by the *CTNNB1* gene; functions in cell adhesion; plays a central role as a key mediator in the Wnt signaling pathway	Nuclear	30–40%	Positive staining supports HCC; not very sensitive, so negative staining is not informative; expressed in the cytoplasm and cell membrane of normal hepatocytes	[38,39,40]
EZH2	Histone methyltransferase; catalytic subunit of PRC2; plays a critical role in the epigenetic regulation of gene expression	Cytoplasmic	80%	Overexpression supports the diagnosis of HCC; expressed in hepatocytes and bile ducts	[42]
Ki67	Nuclear protein associated with cell proliferation; used as a marker for cell proliferation	Nuclear		Most helpful if significantly higher than background liver; not all HCCs show a high proliferation rate	[38,39]
CD34	Transmembrane phosphoglycoprotein; expressed in endothelial cells	Cytoplasmic		Positive staining in HCC sinusoids (capillarization); strong diffuse staining is more common in HCC than in hepatocellular adenoma	[38,39,40]
Reticulin	Constituent of the supporting framework of hepatic parenchyma			Reticulin loss in HCCs; widened cell plates (>2 cells thick)	[1,38,39,40]

^(a)^ Sensitivity may depend on tumor grade and HCC subtypes. Determining the sensitivity and specificity of immunohistochemical markers for HCC will require careful validation in the future. HCC, hepatocellular carcinoma; Hep Par-1, hepatocyte paraffin-1; CEA, carcinoembryonic antigen; EZH2, enhancer of zeste homolog 2; PRC2, polycomb repressive complex 2.

**Table 4 biomedicines-11-02582-t004:** Clinical, pathological, and molecular features of the subtypes of hepatocellular carcinoma ^(a)^.

Subtype	Relative Frequency	Clinical Features	Pathological Features	Molecular Features	Prognosis ^(b)^	References
Steatohepatitic HCC	5–20%	Steatohepatitis due to metabolic syndrome or alcohol abuse can occur in the background liver	>50% of tumor has histological features of steatohepatitis; fat, inflammation, and fibrosis	IL-6–JAK–STAT signaling activation; lower frequency of *CTNNB1*, *TERT* promoter, and *TP53* mutations	Similar	[50,51,52,53,54,55]
Clear cell HCC	3–7%	No distinct findings to date	>80% of tumor shows clear cell morphology due to glycogen accumulation; some steatosis is acceptable	No distinct findings to date	Better	[56,57,58,59,60,61]
Macrotrabecular massive HCC	5%	High serum α-fetoprotein	>50% of tumor shows macrotrabecular growth pattern (>10 cells thick); vascular invasion common	*TP53* mutations and *FGF19* amplifications	Worse	[62,63,64]
Scirrhous HCC	4%	Often mimics cholangiocarcinoma on imaging	>50% of tumor shows dense intratumoral fibrosis	*TSC1/2* mutations; TGF-β signaling activation	Variable, no consensus in the literature	[65,66,67,68,69,70,71,72,73]
Chromophobe HCC	3%	No distinct findings to date	Light, eosinophilic to clear (chromophobic) cytoplasm; mainly bland tumor nuclei, but scattered foci of more conspicuous nuclear atypia; microscopic pseudocysts	Alternative lengthening of telomeres	Similar	[74]
Fibrolamellar carcinoma	1%	Young median age (25 years); no background liver disease	Large eosinophilic tumor cells with prominent nucleoli; dense intratumoral fibrosis with lamellar pattern	Activation of PKA via a *DNAJB1::PRKACA* fusion gene	Similar to HCC in non-cirrhotic liver	[75,76,77,78,79,80,81,82]
Neutrophil-rich HCC	<1%	Elevated serum white blood cell count, C-reactive protein, and IL-6	Numerous and diffuse neutrophils within tumor; may have sarcomatoid areas	Tumor produces G-CSF	Worse	[83,84,85,86,87,88]
Lymphocyte-rich HCC	<1%	No distinct findings to date	Lymphocytes outnumber tumor cells in most areas	No distinct findings to date; not EBV-related	Better	[89,90,91]

^(a)^ The subtypes of HCC are based on the 2019 WHO classification (5th edition) of liver tumors [1]. ^(b)^ Compared to the prognosis of conventional HCC. HCC, hepatocellular carcinoma; IL-6, interleukin-6; JAK, Janus kinase; STAT, signal transducer and activator of transcription; *TERT*, telomerase reverse transcriptase; *FGF19*, fibroblast growth factor 19; *TSC1/2*, tuberous sclerosis complex 1/2; TGF-β, transforming growth factor-β; PKA, protein kinase A; IL-6, interleukin-6; G-CSF, granulocyte-colony stimulating factor; EBV, Epstein–Barr virus.

**Table 5 biomedicines-11-02582-t005:** Clinical, pathological, and molecular features of the provisional subtypes of hepatocellular carcinoma ^(a)^.

Subtype	Relative Frequency	Clinical Features	Pathological Features	Molecular Features	Prognosis ^(b)^	References
*CTNNB1*-mutated HCC	<1%	Enriched for the man; less likely to respond to ICI therapies	Well differentiated with thin trabecular pattern; prominent pseudoglands; bile production	*CTNNB1* mutation	Better	[115,116,117,118]
Sarcomatoid HCC	<1%	Similar risk factors to conventional HCC; can be induced by anticancer therapy, such as chemoembolization	Both the HCC component and major ≥10% spindle cell component lacking specific differentiation	No distinct findings to date	Worse	[119,120]
Lipid-rich HCC	<1%	Non-alcoholic fatty liver disease in the background liver (subset)	Tumor cells have abundant cytoplasm filled with numerous small lipid droplets, resembling microvesicular steatosis	No distinct findings to date	Unclear	[121,122,123]
Myxoid HCC	<1%	Unique radiological findings	Tumor sinusoids are dissected by abundant myxoid material	No distinct findings to date	Unclear	[124,125]
HCC with syncytial giant cells	<1%	No distinct findings to date	Distinctive syncytial multinucleated giant cells	No distinct findings to date	Unclear	[126]
*BAP1* mutated and PKA activated HCC	<1%	Elderly patients, often women; no underlying liver disease	Mixed histological features of HCC and fibrolamellar carcinoma; abundant fibrous stroma	*BAP1* gene mutation	Worse	[127]
Transitional liver cell tumor	<1%	Older children and adolescents; elevated serum α-fetoprotein	Tumor cells are well to moderately differentiated; focal areas resembling hepatoblastomas	No distinct findings to date	Similar	[128]
Cirrhotomimetic HCC	<1%	Background liver cirrhosis in most cases; tumor burden can be underestimated by imaging studies	Numerous tumor nodules (>30); tumor cells are well to moderately differentiated	No distinct findings to date	Worse	[129,130,131,132]
Progenitor HCC	<5%	Increased serum α-fetoprotein	Morphologically compatible with HCC but >5% of tumor express cytokeratin 19; infiltrative growth, large tumors, frequent vascular invasion, poor differentiation	*TP53* mutation; chromosomal instability	Worse	[133,134,135,136,137,138,139]

^(a)^ The provisional subtypes of HCC are not recognized in the 2019 WHO classification (5th edition) of the liver tumors. ^(b)^ Compared to the prognosis of conventional HCC. HCC, hepatocellular carcinoma; ICI; immune checkpoint inhibitor; PKA, protein kinase A.

## Data Availability

All data were included in the manuscript.

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
