# Peer review of "Advances in Histological and Molecular Classification of Hepatocellular Carcinoma"

_biomedicines, 2023, doi:10.3390/biomedicines11092582_

Round 1

Reviewer 1 Report

The manuscript is fairly well written and needs careful revision by a native English speaker to easily deliver the message in a better way

Author Response

<Reviewer 1>

The manuscript authored by Choi et al and entitled "Advances in Histological and Molecular Classification of Hepatocellular Carcinoma" provides insights and updates on HCC etiology, pathogenesis in addition to its molecular genetics, immnunohistopathological and clinical features. Although the manuscript is well structured with detailed information on the previously mentioned aspects, the manuscript is missing updates son recent non-invasive biomarkers for HCC diagnosis that have attracted researchers’ interest over the elapsed few years. The authors need to give more details on the importance of using non-invasive biomarkers such as circulating tumor DNA, circulating tumor cells, extracellular vesicles, long non coding RNAs (lncRNAs), micro RNAs (miRNAs) and circulating amino acids, etc. Also authors need to briefly discuss current and promising therapeutic approaches/interventions to enrich the manuscript. By doing so the manuscript might be an asset to the currently available databases concerned with HCC pathogenesis, diagnosis and therapeutic intervention. The manuscript is fairly well written and needs careful revision by a native English speaker to easily deliver the message in a better way. The experimental design seems to be relevant and consistent, however various non-invasive biomarkers are missing and should be detailed in this review to be inclusive and comprehensive. The conclusions are well supported. Moreover, the provided figures (particularly figure 1 to fig 5) along with the tables should cite the relevant reference numbers. I believe the current version of the manuscript needs critical revision in addition to addressing the below concern and to comply with all the typos detailed below. The authors also need to address some major concerns and comments prior to accepting this paper in order to be beneficial to the wide readership of the prestigious journal of Biomedicines. I would therefore grant publication of this article in Biomedicines after considering the following major points;

Raised Comments;

- It is not clear whether the 5 figures provided (from figure 1 to figure 5) are derived from authors’ work or not??? IF so, then this should be clearly indicated. If not, then full credit and citation of references/resources should be clearly stated.

-> Thank you for your comments. Figures 1, 2, 3, 4, and 5 were derived from our working files and have not been previously published.

- The three tables provided (from table 3 to table 5) should be supplied with reference numbers in the last column to the right of the provided table.

-> Thank you for your comments. Reference numbers are given the last column to the right of Table 3, 4, and 5.

- It would be ideal to include a figure depicting liver deterioration from healthy liver, to fibrosis, cirrhosis and then HCC with highlights on the causing factors for HCC (ideally this should be figure 1)

-> Thank for your suggestion. We have included a figure showing the development of HCC.

“Figure 1. The diagram illustrates the development of hepatocellular carcinoma (HCC) from a healthy liver. Major etiological factors include HBV, HCV, NAFLD, and alcohol. HCC can arise within a dysplastic nodule in cirrhosis through a multistep process or develops de novo in a non-cirrhotic liver. HBV, hepatitis B virus; HCV, hepatitis C virus; NAFLD, non-alcoholic fatty liver disease.”

- The manuscript is well structured with detailed information on the previously mentioned aspects, the manuscript is missing updates on the importance of using non-invasive biomarkers such as circulating tumor DNA (ctDNA) with DNA abberations in addition to DNA integrity (ratio of ctDNA length/ whole circulating DNA length) .... Please cite the following references * Non-Invasive Biomarkers for Immunotherapy in Patients with Hepatocellular Carcinoma: Current Knowledge and Future Perspectives. Cancers 2022, 14(19), 4631; https://doi.org/10.3390/cancers14194631.

->We appreciate your suggestion, We have cited the journal Cancers 2022,14,4631 and described non-invasive biomarkers as follow.

“In recent years, non-invasive biomarkers such as circulating tumor DNAs (ctDNAs) and extracellular vesicles (EVs) have been widely used in oncology research [186]. ctDNA are DNA fragments that originate from necrotic or apoptotic cells and can be detected in peripheral blood. ctDNA, characterized by DNA aberrations and evaluated by DNA integrity (the ratio of ctDNA to the whole circulating DNA length), can serve as a biomarker for early diagnosis, therapeutic monitoring, and prediction of tumor recurrence in HCC [187,188]”

  1. Pallozzi, M.; Di Tommaso, N.; Maccauro, V.; Santopaolo, F.; Gasbarrini, A.; Ponziani, F.R.; Pompili, M. Non-invasive biomarkers for immunotherapy in patients with hepatocellular carcinoma: current knowledge and future perspectives. Cancers (Basel) 2022, 14. doi:10.3390/cancers14194631.
  2. Wu, X.; Li, J.; Gassa, A.; Buchner, D.; Alakus, H.; Dong, Q.; Ren, N.; Liu, M.; Odenthal, M.; Stippel, D. et al. Circulating tumor DNA as an emerging liquid biopsy biomarker for early diagnosis and therapeutic monitoring in hepatocellular carcinoma. Int. J. Biol. Sci. 2020, 16, 1551-1562. doi:10.7150/ijbs.44024.
  3. Wang, J.; Huang, A.; Wang, Y.P.; Yin, Y.; Fu, P.Y.; Zhang, X.; Zhou, J. Circulating tumor DNA correlates with microvascular invasion and predicts tumor recurrence of hepatocellular carcinoma. Ann. Transl. Med. 2020, 8, 237. doi:10.21037/atm.2019.12.154.

- The manuscript is missing updates on the importance of using non-invasive biomarkers such as extracellular vesicles.

->We appreciate your suggestion. We have described extracellular vesicles as follows.

“EVs are lipid bilayer-enclosed particles released by almost all cell types and can be categorized into exosomes and microvesicles based on their size. They play a role in intercellular communication. For example, EVs mediate interactions between tumor cells, immune cells, and other components of the tumor microenvironment in HCC [189]. Tumor-derived exosomes may be involved in resistance to the chemotherapeutic agents in HCC [190]. Consequently, EVs can be potential biomarkers and therapeutic targets in HCC.

  1. Li, S.; Chen, L. Exosomes in pathogenesis, diagnosis, and treatment of hepatocellular carcinoma. Front. Oncol. 2022, 12, 793432. doi:10.3389/fonc.2022.793432.
  2. Wang, H.; Yu, L.; Huang, P.; Zhou, Y.; Zheng, W.; Meng, N.; He, R.; Xu, Y.; Keong, T.S.; Cui, Y. Tumor-associated Exosomes Are Involved in Hepatocellular Carcinoma Tumorigenesis, Diagnosis, and Treatment. J. Clin. Transl. Hepatol. 2022, 10, 496-508. doi:10.14218/jcth.2021.00425.

- The manuscript is missing updates on the importance of using non-invasive biomarkers such as micro RNAs (miRNAs)

* Circulatory miR-221 & mR-542 expression profiles as potential molecular biomarkers in Hepatitis C virus mediated liver cirrhosis and hepatocellular carcinoma. Virus Res. 2021; 296, 198341.

* Investigating circulatory microRNA expression profiles in Egyptian patients infected with hepatitis C virus mediated hepatic disorders. Meta Gene; 2020; 26, 100792.

-> We appreciate your suggestion. We quoted the journal Virus Res. 2021;296, 198341 and Meta Gene 2020;26, 100792 and described as follows.

“MicroRNAs (miRNAs) are a class of small non-coding RNAs, usually 19–25 nucleotides in length, that play a crucial role in regulating gene expression within cells. For example, circulatory miR-542 is significantly upregulated in HCV-associated cirrhosis and chronic HCV groups but is markedly downregulated in HCV-associated HCC groups [191]. In contrast, circulatory miR-21 and miR-122 are significantly upregulated in HCC groups compared to healthy and cirrhotic groups [192]. Circulatory miRNAs are increasingly recognized as potential non-invasive biomarkers for HCC.”

  1. Yasser, M.B.; Abdellatif, M.; Emad, E.; Jafer, A.; Ahmed, S.; Nageb, L.; Abdelshafy, H.; Al-Anany, A.M.; Al-Arab, M.A.E.; Gibriel, A.A. Circulatory miR-221 & miR-542 expression profiles as potential molecular biomarkers in Hepatitis C Virus mediated liver cirrhosis and hepatocellular carcinoma. Virus Res. 2021, 296, 198341. doi:10.1016/j.virusres.2021.198341.
  2. Gibriel, A.A.; Al-Anany, A.M.; Al-Arab, M.A.E.; Azzazy, H.M.E. Investigating circulatory microRNA expression profiles in Egyptian patients infected with hepatitis C virus mediated hepatic disorders. Meta. Gene 2020, 26. doi:10.1016/j.mgene.2020.100792.

- Also the authors need to discuss some useful molecular techniques such as NGS and microarray platforms in detection of nucleic acids and DNA/RNA fragments....Please cite the

following references

* Novel pathogenic mutations and further evidence for clinical relevance of genes and variants causing hearing impairment in Tunisian population. Journal of Advanced Research, 2021, 31, pp. 13–24.

-> We appreciate your suggestion. We cited the Journal of Advanced Research 2021,31, 13-25 and described as follows.

“Some molecular techniques, such as next-generation sequencing (NGS) and microarray platforms, are useful for detecting alterations in nucleic acids, including DNA and RNA [197]. These advanced technologies will significantly contribute to our understanding of the biology of HCC, refining its classification, improving prognostication, and guiding therapeutic strategies.”

  1. Souissi, A.; Ben Said, M.; Ben Ayed, I.; Elloumi, I.; Bouzid, A.; Mosrati, M.A.; Hasnaoui, M.; Belcadhi, M.; Idriss, N.; Kamoun, H. et al. Novel pathogenic mutations and further evidence for clinical relevance of genes and variants causing hearing impairment in Tunisian population. J. Adv. Res. 2021, 31, 13-24. doi:10.1016/j.jare.2021.01.005.

- Also authors need to briefly discuss current and promising therapeutic approaches/interventions to enrich the manuscript.

-> Thank you for your comments. We have described current and promising therapeutic approaches as follows.

“HCC is an aggressive tumor. While surgical treatment can offer a curative option for early and resectable HCCs, patients with inoperable advanced HCCs often rely on alternative therapeutic approaches. Current and promising treatments include systemic therapies with multikinase inhibitors such as sorafenib and lenvatinib, molecularly targeted therapies, and immunotherapies, notably immune checkpoint inhibitors [198]. The rapidly evolving therapeutic landscape of advanced HCC warrants further research [199].”

  1. Marzi, L.; Mega, A.; Gitto, S.; Pelizzaro, F.; Seeber, A.; Spizzo, G. Impact and novel perspective of immune checkpoint inhibitors in patients with early and intermediate stage 199. HCC. Cancers (Basel) 2022, 14, 3332. doi:10.3390/cancers14143332. Rimassa, L.; Finn, R.S.; Sangro, B. Combination immunotherapy for hepatocellular carcinoma. J. Hepatol. 2023, 79, 506-515. doi:10.1016/j.jhep.2023.03.003.

- The manuscript is fairly well written and needs careful revision by a native English speaker to easily deliver the message in a better way.

-> Thank you for your comments. We revised the manuscript by a native English speaker.

- Please provide full names for all mentioned targets and genes as they first appear in the

manuscript.

-> Thank you for your comments. We have given full names for all targets and genes when they first appeared.

Reviewer 2 Report

The current manuscript arouses interest for readers and provides an important clue to understanding HCC based on histology, immunology, etiology, and molecular biology. This manuscript has been well written and is easily readable to understand. However, there are some issues that should be altered or modified.

(1)  Subsection 10.5, first para, last sentence: How is the prognosis of HCC-SG clear? Is it good or poor?

(2)  Subsection 10.8, first para: What are the precise biological mechanisms underlying the unusual growth pattern? How is it clear? Does this mean intrahepatic metastases of main portal vein HCC thrombosis?

(3)  Subsection 9.8: Does the abbreviation “LEL-HCC” mean “lymphoepithelioma-like HCC”? If so, the authors should either use the abbreviation or the full spelling.

(4)  Section 7: Similarly, the authors should either use the full spelling or the abbreviation for glutamine synthetase (GS).

(5)  There are some typographic or grammatical errors: e.g., section 6, line 4; section 3, second para, line 2.

Author Response

<Reviewer 2>

The current manuscript arouses interest for readers and provides an important clue to understanding HCC based on histology, immunology, etiology, and molecular biology. This manuscript has been well written and is easily readable to understand. However, there are some issues that should be altered or modified.

 (1)  Subsection 10.5, first para, last sentence: How is the prognosis of HCC-SG clear? Is it good or poor?

-> Thank you for your comments. We have described as follow.                         “The prognosis remains unclear.”

(2)  Subsection 10.8, first para: What are the precise biological mechanisms underlying the unusual growth pattern? How is it clear? Does this mean intrahepatic metastases of main portal vein HCC thrombosis?

-> Thank you for your comments. We have described as follows.                     “The precise biological mechanisms underlying this unusual growth pattern remain unclear.”

(3)  Subsection 9.8: Does the abbreviation “LEL-HCC” mean “lymphoepithelioma-like HCC”? If so, the authors should either use the abbreviation or the full spelling.

-> We replaced “LEL-HCC” with “lymphoepithelioma-like HCC.”

(4)  Section 7: Similarly, the authors should either use the full spelling or the abbreviation for glutamine synthetase (GS).

-> Thank you for your comments. We replaced “glutamine synthetase (GS)” with “glutamine synthetase” in the text.

 (5)  There are some typographic or grammatical errors: e.g., section 6, line 4; section 3, second para, line 2.

-> Thank you for your comments. We changed as follows.

section 6, line 4; at IGF2 gene have been identified in patients with HCC [29,30] -> “in the IGF2 (insulin-like growth factor 2) gene were identified in patients with HCC [29,30]”

section 3, second para, line 2.: “sequence of hepatocytes death sequence”-> sequence of hepatocytes death

Reviewer 3 Report

As the authors indicate, this review provides an update on HCC pathology, focusing on molecular genetics, histological subtypes, and diagnostic approaches. Some points should be noted as below.

1) Artificial intelligence and digital pathology for HCC have gained many advances in recent years. The authors also wrote that, ..will likely be effective methods for HCC diagnosis, classification, risk stratification, treatment planning, and treatment outcome prediction [190,191]”, it sounds interesting. I suggest it should be better listed as a subtopic in order to present a more detail for the readers.

2) About the pathogenesis of HCC, and 10.2. Sarcomatoid HCC section, such sarcomatoid (spindle cells) pathological changes are indeed found in many HCC tissues clinically, just in different proportion. A recent paper proposes that the cancer is an ecological disease: a multidimensional spatiotemporal "unity of ecology and evolution" pathological ecosystem (https://www.thno.org/v13p1607.htm). In this paper, the author also proposes that this cancer subtype is reflection of morphological adaptation to the external environment. It should be interesting to discuss about it.

3) The size of images in Figure 1 and 2 were not identical. A, B and C in all of images should be labeled. 

Author Response

<Reviewer 3>

As the authors indicate, this review provides an update on HCC pathology, focusing on molecular genetics, histological subtypes, and diagnostic approaches. Some points should be noted as below.

1) Artificial intelligence and digital pathology for HCC have gained many advances in recent years. The authors also wrote that, “..will likely be effective methods for HCC diagnosis, classification, risk stratification, treatment planning, and treatment outcome prediction [190,191]”, it sounds interesting. I suggest it should be better listed as a subtopic in order to present a more detail for the readers.

-> Thank you for your comments. We have described artificial intelligence for HCC in a new paragraph as follows.

“Artificial intelligence (AI) represents a form of machine intelligence, encompassing computational search algorithms, machine learning, and deep learning models [190]. Deep learning, for instance, has demonstrated its capability to predict RNA-Seq expression of tumors from whole-slide images [191]. Such models are increasingly utilized to classify cancer types and identify gene mutations [Coudray]. AI combined with digital pathology is poised to become an effective tool for HCC diagnosis, classification, risk assessment, treatment planning, and prediction of treatment outcomes [190,191]. However, further research is still needed to standardize AI algorithms and evaluate them rigorously through prospective studies. This will ultimately enhance their interpretability, generalizability, and transparency [190, Pellat].”

Coudray, N.; Ocampo, P.S.; Sakellaropoulos, T.; Narula, N.; Snuderl, M.; Fenyö, D.; Moreira, A.L.; Razavian, N.; Tsirigos, A. Classification and mutation prediction from non-small cell lung cancer histopathology images using deep learning. Nat. Med. 2018, 24, 1559-1567. doi:10.1038/s41591-018-0177-5.

Pellat, A.; Barat, M.; Coriat, R.; Soyer, P.; Dohan, A. Artificial intelligence: A review of current applications in hepatocellular carcinoma imaging. Diagn. Interv. Imaging 2023, 104, 24-36. doi:10.1016/j.diii.2022.10.001.

2) About the pathogenesis of HCC, and “10.2. Sarcomatoid HCC section”, such sarcomatoid (spindle cells) pathological changes are indeed found in many HCC tissues clinically, just in different proportion. A recent paper proposes that the cancer is an ecological disease: a multidimensional spatiotemporal "unity of ecology and evolution" pathological ecosystem (https://www.thno.org/v13p1607.htm). In this paper, the author also proposes that this cancer subtype is reflection of morphological adaptation to the external environment. It should be interesting to discuss about it.

-> We appreciate your suggestion. We have described as follows.

“Recently, it has been suggested that the sarcomatoid changes in carcinoma cells might reflect the morphological and biological adaptation to the external environments [Luo]. Furthermore, cancer can be viewed as an ecological disease involving a multidimensional spatiotemporal ecological and evolutionary process.”

Luo, W. Nasopharyngeal carcinoma ecology theory: cancer as multidimensional spatiotemporal "unity of ecology and evolution" pathological ecosystem. Theranostics 2023, 13, 1607-1631. doi:10.7150/thno.82690.

3) The size of images in Figure 1 and 2 were not identical. A, B and C in all of images should be labeled.

-> Thank you for your comments. Figure 1 and 2 has been resized. A, B, and C in all images have been labeled.

Round 2

Reviewer 1 Report

I believe by now, authors have responded positively to most of the raised comments and the manuscript now looks better and would be suitable to the wide readership of the prestigious journal of Biomedicines.  

Author Response

I revised the manuscript according to the reviewers' comments.

Reviewer 3 Report

The authors have fully answered my questions.

Author Response

(The authors gave the same response as above.)
